# Base excision repair and double strand break repair cooperate to modulate the formation of unrepaired double strand breaks in mouse brain

Aris A. Polyzos[1] ✉, Ana Cheong [2], Jung Hyun Yoo[1], Lana Blagec[1], Sneh M. Toprani[2], Zachary D. Nagel [2] & Cynthia T. McMurray [1] ✉

We lack the fundamental information needed to understand how DNA damage in the brain is generated and how it is controlled over a lifetime in the absence of replication check points. To address these questions, here, we integrate cell-type and region-specific features of DNA repair activity in the normal brain. The brain has the same repair proteins as other tissues, but normal, canonical repair activity is unequal and is characterized by high base excision repair (BER) and low double strand break repair (DSBR). The natural imbalance creates conditions where single strand breaks (SSBs) can convert to double strand breaks (DSBs) and reversibly switch between states in response to oxidation both in vivo and in vitro. Our data suggest that, in a normal background of repair, SSBs and DSBs are in an equilibrium which is pushed or pulled by metabolic state. Interconversion of SSB to DSBs provides a physiological check point, which would allow the formation of unrepaired DSBs for productive functions, but would also restrict them from exceeding tolerable limits.

Terminally differentiated cells such as neurons need efficient DNA repair mechanisms to maintain their genomic integrity for decades[1-4]. However, the repair mechanisms in brain cells are often inferred from those of cancer cells, which differ substantially. Cancer cells are mitotic, often adapt to a unique immune microenvironment in vivo[1], or harbor alterations in DNA repair pathways. Homogenous clonal populations in established cancer cell lines[5,6] often have the same repair capacity in all cells. In contrast, the brain is highly heterogenous with regions differing in cell composition, metabolism, and tissue structure[7,8]. There is no cell division in adult neurons, and, while glial cells have the potential to divide, they are mainly dormant in the absence of insult[9,10]. Without replication, brain cells lack the cell cycle check points that guard DNA integrity in dividing cells. Thus, neurons need alternative means to avoid dysfunction from DNA damage, but little is known about DNA repair mechanisms in the normal brain or how damage is kept in check over a lifetime.

In most cases, the healthy brain is well protected by the cranium from external insults such as sunlight, and the blood–brain barrier prevents against some damage from systemic exposures such as drugs or infectious agents[11,12]. Thus, it is generally accepted that DNA damage in the brain is dominated by endogenous base damage including deamination, alkylation, or oxidation of individual DNA bases[13-20]. These lesions are typically repaired through well-understood pathways of base excision repair (BER)[17-20]. For example, a common form of DNA base damage, 7-hydro-8-oxo-2′-deoxyguanosine (8-oxo-dG)[18,19], arises from reactive oxygen species (ROS) as a biproduct of mitochondrial respiration[14,20]. Removal of the oxidized guanine by a glycosylase creates a widowed apurinic site opposite cytosine. Apurinic/Apyrimidinic endonuclease 1 (APE1) nicks the phosphodiester backbone, leaving a transient single-strand break (SSB) as an intermediate during the repair process[13,19]. Removal of bulkier oxidative lesions such as purine 5′,

[1]Division of Molecular Biophysics and Integrated Bioimaging, Lawrence Berkeley National Laboratory, Berkeley, CA, USA. [2]Department of Environmental Health, John B Little Centre for Radiation Sciences, Harvard T.H. Chan School of Public Health, Boston, MA, USA. ✉e-mail: aapolyzos@lbl.gov; ctmcmurray@lbl.gov

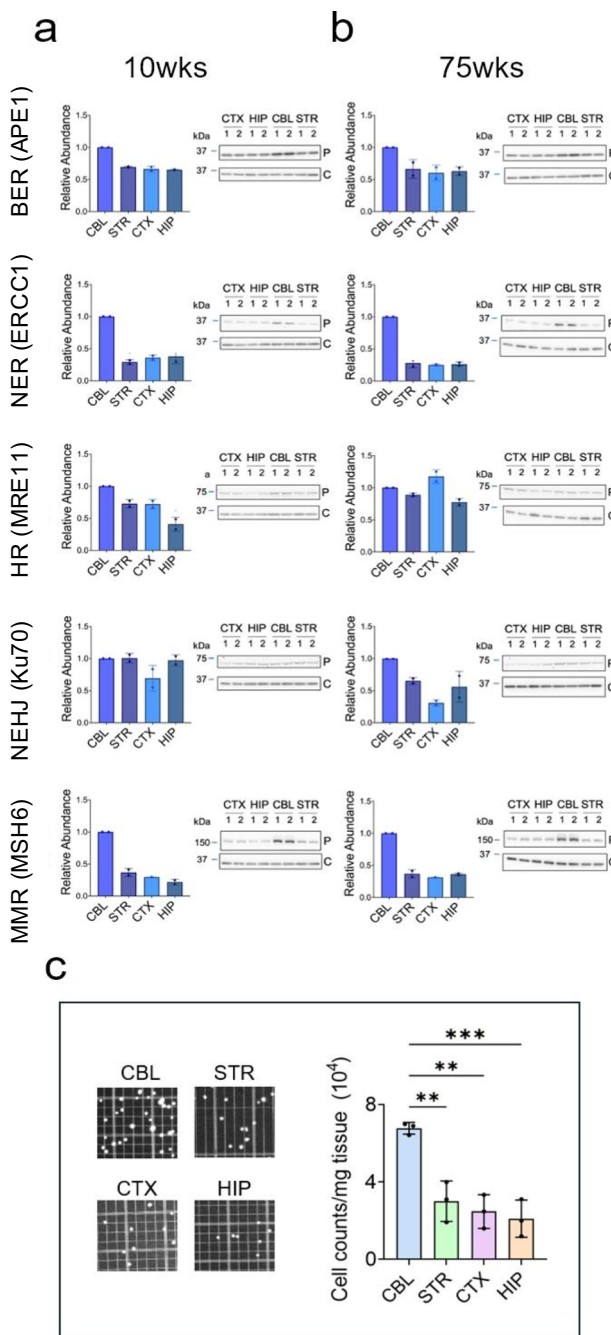

**Fig. 1 | Mouse brain expresses the machinery to carry out DNA repair.** Expression of DNA repair machinery was determined from four brain regions: the Cerebellum (CBL), Striatum (STR), Cortex (CTX), and Hippocampus (HIP) of 10–11 weeks (**a**) and 75 weeks (**b**) in C57BL/6J male mice. The C57BL/6J strain is clonal, and genetically identical animals have similar protein profiles. We selected 4 genetically identical animals from the colony, $n = 2$ young animals of 10 weeks and $n = 2$ animals of 75 weeks. At each age, the extracts from the two each animal were run side by side, indicated by the number 1 and 2 in the gel. Eight technical replicates of the SDS-PAGE gels were analyzed. Five technical replicate sets of SDS-PAGE gels were analyzed in Fig. 1 and three additional replicate sets were included in Supplementary Fig. 1. Each replicate gel was transferred to membranes and probed with specific antibodies to a representative protein (P) or GAPDH (C), shown to the side of each plot (APE1); Excision Repair Cross-complementation group 1 (ERCC1); Meiotic Recombination 11 Homolog 1 (MRE11); X-ray repair cross-complementing 6 (Ku70); MutS homolog 6 (MSH6). Protein antibodies are listed in Supplementary Table 2. The expression of DNA repair proteins in each region was plotted relative to the CBL. Error bars represent minimum (lower bar) and maximum (upper bar) values for the two clonal samples ($n = 2$) at each age and were similar. Full uncropped gels are provided in Supplementary Source file. **c** Hemocytometer image of cell number (left) and plot of cell count per mg tissue (right) from four dissected brain regions. Bars represent the standard error of the mean (SEM) with minima (lower bar) and maxima (upper bar) indicated. Cell number was determined from brains of $n = 3$ animals; the average of five random hemocytometer fields per animal are plotted. The regional variations in cell number per mg protein were statistically significant (one-way ANOVA). $P$ is *$0.01 < P \le 0.01$, **$0.001 < P \le 0.01$, ***$0.0001 < P \le 0.001$. Source data are provided in Source data file.

transcription of an essential gene[39]. Furthermore, there is emerging evidence that DSBs act as key regulatory elements for stimulating gene expression and serve adaptive roles[40–45]. For example, stimulation of neuronal activity in animals during exercise results in the transient expression of a small subset of early response genes (*Fos, FosB, Npas4,* and *Egr*1) in a manner that depends on the induction of DSBs[40–45]. Thus, DSBs can stimulate gene expression in the context of normal physiology and help rather than hurt cellular function[40–45]. However, we lack the fundamental information needed to understand how DSBs are checked in normal, non-dividing cells over time.

To address this gap, we have measured the DNA repair capacity, the repair protein expression profiles and DSB levels among brain regions in healthy mice. We show here that the DNA repair machinery in the brain is like that of other tissues. However, the number of DSBs does not solely depend on DSBR. Rather, the activity of DNA repair pathways is coordinated to modulate the formation of DSBs by curbing their conversion from SSBs. The pathways that generate DSBs in the brain are as important as those that repair them and, together, they keep DSBs within acceptable limits in the absence of replication check points.

## Results

### Expression of the DNA repair machinery is region specific
The potential for DNA repair in the brain depends in part on the expression level of the repair machinery. Thus, we tested whether machinery from major DNA repair pathways could be detected in four regions of the brain from *C57BL/6J* mice: the cerebral cortex (CTX), the hippocampus (HIP), the cerebellum (CBL) and the striatum (STR) at young (around 10 weeks) (Fig. 1a) and at old ages (75 weeks) (Fig. 1b). Proteins from five major DNA repair pathways were evaluated in each region (Fig. 1 and Supplementary Fig. 1). These included BER, NER/TCR, mismatch repair (MMR), as well as HR (Homologous Recombination) and NHEJ (Non-Homologous End Joining) pathways for DSBR[13,17]. Due to the complex and multi-component nature of repair complexes, we did not evaluate all repair proteins in each pathway. Rather, we tested the expression of one or two key proteins in each pathway (Fig. 1 and Supplementary Fig. 1).

The *C57BL/6J* strain is clonal and the animals are genetically identical with the same protein profiles. Thus, four male animals from

8-deoxynucleosides generates gapped SSBs during nucleotide excision repair in the global genome (GG-NER)[21–23] or by more specialized transcription-coupled NER (TC-NER or TCR)[24,25] in an actively transcribed gene. NER can cooperate with BER[26–29] in the repair of endogenous damage but is typically reserved in cases where BER is not sufficient.

The processing of double-strand breaks (DBSs) in non-dividing cells, however, remains enigmatic. There is a widely held view that DNA DSBs in neurons are toxic. Indeed, deficiencies in double-strand break repair (DSBR), which causes cancer in dividing cells, lead to DSBs and death in the brain[30,31]. Furthermore, DSBs are implicated in age-related neurodegenerative diseases, such as Alzheimer's disease (AD)[32–34], Parkinson disease (PD)[35–37], and Amyotrophic lateral sclerosis (ALS)[38]. Whether DSBs are cytotoxic under normal physiological conditions, however, remains unclear. In the absence of replication, DSBs can be tolerated in non-dividing cells unless there is interference with

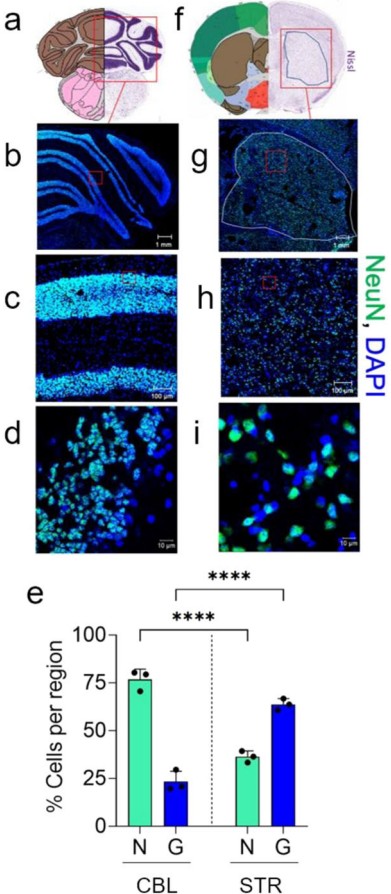

**Fig. 2 | The relative proportion of neurons and glia vary among brain regions.** Mouse Allen Mouse Brain Atlas map[46] highlighting coronal brain sections of the CBL (**a**) or the STR (**f**) (Allen Mouse Brain Atlas; mouse.brain-map.org/shown/2011 coronal reference atlas) (http://atlas.brain-map.org/atlas?atlas=1). The anatomical annotations from the Allen Mouse Brain Atlas for STR and CBL were used at the same slice positions. IF staining of neurons and glial cells from coronal brain sections of male (7 weeks) C57BL/6J mice in the CBL (**b**–**d**) or the caudoputamen of the STR (**g**–**i**). Neurons are co-labeled with an anti-NeuN antibody (green) and DAPI (blue) and appear as teal. Glia stain only with DAPI and appear blue. Red boxes illustrate a segment of the coronal brain section that was magnified in the image below it; scale bars are 1 mm (**b**, **g**), 100 μm (**c**, **h**), and 10 μm (**d**, **i**), as indicated. **e** The proportion of each cell type in $n = 50$ cells were assessed in each region from three random tissue sections ($n = 3$). The data were plotted as % of total cells counted per region in ($n = 3$) animals. The total % neuronal (N, green) and % glial (G, blue) cell content was visualized to directly compare the CBL and STR. Bars are standard error of the mean (SEM). Source data are provided in Source data file. Regional cell type comparison was determined by a one-way ANOVA, ****$P < 0.0001$.

the colony were chosen to establish whether DNA repair machinery was expressed in the mouse strain. Extracts from brain regions of two young animals (10–11 weeks) and two old animals (75 weeks) were resolved on SDS-PAGE gels and proteins were detected using specific antibodies (Fig. 1 and Supplementary Fig. 1). As judged by antibody staining, all brain regions of young (10–11 weeks) (Fig. 1a and Supplementary Fig. 1a) and old (75 weeks) animals (Fig. 1b and Supplementary Fig. 1b) expressed the representative DNA repair proteins from all five pathways. As expected, little variation in expression of DNA repair proteins was observed in the two genetically identical animals in each age group (Fig. 1a, b, error bars are the minimum (lower bar) and maximum (upper bar) for $n = 2$ samples). Brain regions, however, differed in cell density, cell type proportion, and cell size (see Fig. 2a, f, shown is Allen Brain Atlas Map[46] of CBL and STR), which can change nuclear protein concentration[47] (Fig. 2), and precluded quantitative

regional comparisons by western analysis. The CBL, for example, was an outlier. Although normalized to total protein for loading, the cerebellar cells were small and had five times more cells per mg tissue relative to other regions (Fig. 1c). When the samples were adjusted for cell number, the protein expression level of the CBL was similar or slightly lower than in other regions. The CBL also differed in neuronal and glial cell content relative to the other brain regions (Fig. 2). For example, we co-stained tissue sections from the CBL (Fig. 2a) and the STR (Fig. 2f) with the neuron-specific marker NeuN (green)[48] together with nuclear DAPI (Blue). Neurons stain with both NeuN and DAPI and appear teal in the overlay (N in Fig. 2b–d, g–i). The NeuN antibody does not label glia (G), which stained only with DAPI and appear as blue (G, Fig. 2b–d, g–i). Indeed, the CBL and STR differed significantly in their cell type proportions (Fig. 2e). The dense layers of the CBL were roughly 80% neurons (Fig. 2c, d, teal), which were separated by the lighter, more diffuse glial segments comprising 20% of the cells (Fig. 2c, d, diffuse blue). In contrast, the STR was only 20% neurons, which were randomly arranged among glia throughout the region (Fig. 2g–i). Thus, western analysis confirmed that each brain region expressed the proteins required to carry out DNA repair, but quantitative comparisons of regional protein expression required cell type resolution (Fig. 3).

## DNA repair machinery in the brain is expressed at a higher level in neurons

The cell type distribution of DNA repair proteins was measured at the single cell level in whole brain tissue sections (Fig. 3, Supplementary Figs. 2 and 3). Each section was co-stained with DAPI nuclear stain, the neuron-specific marker NeuN[48] and an antibody to a designated pathway protein to determine its localization in neurons or glia (Supplementary Fig. 2a, b). Overlays of three antibody signals often obscured color resolution (Supplementary Fig. 2a, D/N/M in image 1 in each row), thus each antibody or stain channel was also included as an individual image (Supplementary Fig. 2a, images 2–4 in each row). Expression was assigned as neuronal if the IF intensity of the expressed protein occurred in NeuN (+) cells (Supplementary Fig. 2a) and as glial if the IF intensity of the expressed protein occurred in NeuN (−) cells (Supplementary Fig. 2b). For example, in all four brain regions, MSH6 co-stained with NeuN (+) neurons (Supplementary Fig. 2a), while staining was weak in NeuN (−) glia (Supplementary Fig. 2b). All DNA repair proteins had the same staining pattern (Supplementary Fig. 2c, d). The per cell protein expression was determined from the average IF intensity of the protein from 50 neurons (NeuN (+) cells) and 50 glia (NeuN (−) cells) (Fig. 3 and Supplementary Fig. 3), which were randomly selected in each brain region after NeuN staining (Fig. 3a). IF had the advantage that quantification was internally controlled, i.e., cell type protein expression in single cells was measured for all proteins side-by-side in the same brain section. Indeed, the expression of the representative repair proteins from all five pathways was highly cell-type dependent (Fig. 3b, c). The highest repair protein expression was detected in NeuN (+) cells, indicating that they were neurons (Supplementary Fig. 2c). Robust expression of DNA repair machinery appeared to be an inherent property of neurons as they expressed the highest protein level in both young (around 10 weeks) (Fig. 3b, c) and in old (75 weeks) (Supplementary Fig. 3) animals in all the brain regions. Glia generally expressed the same DNA repair proteins as neurons, but at lower levels (Fig. 3) (Supplementary Fig. 3). Indeed, the neuronal/glial ratio (N/G) of DNA repair protein expression fell between 2:1 and 4:1 for NER (XPA), HR (MRE11), NHEJ (Ku80) and MMR (MSH6), but was as high as 10:1 for APE1 of the BER pathway (Supplementary Fig. 4) and did not change significantly with age (compare Fig. 3 and Supplementary Fig. 3). Single cell quantification by antibody staining is more variable relative to western analysis. Nonetheless, over multiple brain segments, the region and cell type-specific content in repair protein expression was consistent. DNA repair machinery was

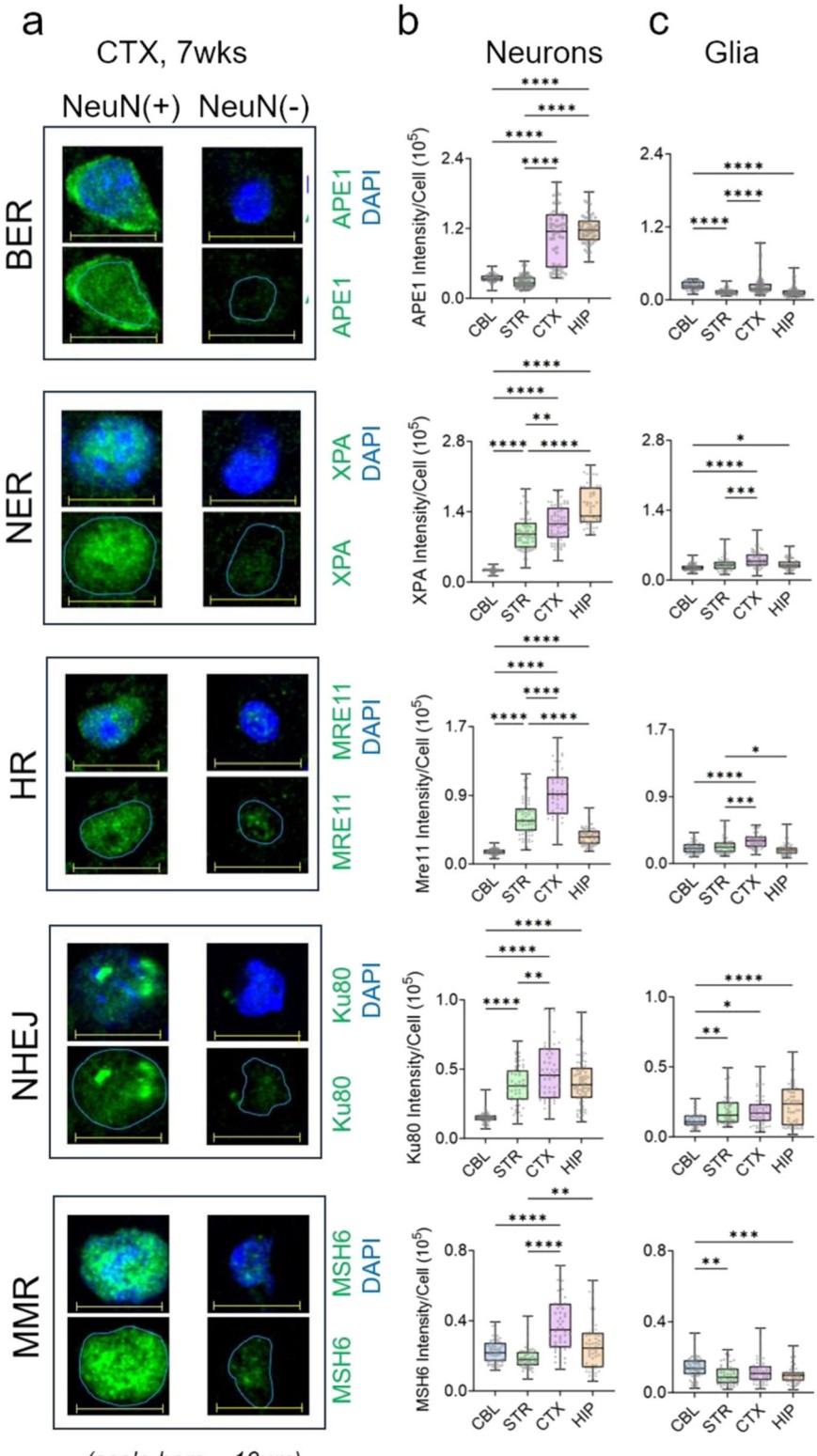

**Fig. 3 | Expression of the DNA repair machinery in the mouse brain is region and cell type specific. a** The quadrant of cell images shows expression of two representative cells for DNA repair proteins of five pathway in NeuN (+) (neurons) and NeuN (−) cells (glia) ± DAPI from cortex of 7 weeks male mice, as indicated. Scale bar is 10 μm. Three additional images for protein expression of each DNA repair protein in NeuN (+) (neurons) and NeuN (−) cells (glia) ± DAPI are included in Supplementary Fig. 2. The nuclear contours from the DAPI stain outlined in light blue indicate the position of the nucleus and highlights the poor protein staining intensity in glia. **b**, **c** The average per cell expression was quantified from IF intensity for each protein in 50 neurons and 50 glia randomly selected in tissue sections from four brain regions: (CBL (blue), STR (green), CTX (pink), HIP (orange) ($n = 3$ animals). The repair proteins quantified were APE1, XPA, MRE11, Ku70, and MSH6, as indicated. Data are displayed as a box and whisker plot, where a box indicating the 25th to 75th percentile values, the line indicates the median, with the lower whiskers representing the minimum 25% of data values and the whiskers above the box representing the 25% maximum values. Statistical probability of variance ($P$) was calculated from $n = 50$ cells of each type (Neuron (Neu), and (GL) Glia) using two-tailed homoscedastic t-test. Statistical significance among regional expression comparisons is tabulated in Supplementary Table 1. *$P = < 0.05$, **$P < 0.005$, ***$P < 0.0005$, ****$P < 0.00005$.

compartmentalized in the brain, i.e., the expression of DNA repair proteins varied not only among brain regions but also within regions, depending on their cell type composition.

## DSBs form despite expression of the DSBR machinery

The expression patterns of the DNA repair machinery suggested that neurons were well equipped to repair damage, using the same repair machinery as found in other tissues. However, the striking cell type-specific expression of the DNA repair machinery raised the issue as to whether DNA damage in the brain was also compartmentalized. Neurons might not accumulate lesions to the same extent as glia due to abundant repair machinery. To address this question, we followed DSB formation over time and quantified it in neurons and glia within each brain region using IF in whole tissue sections (Fig. 4). DSBs are easily visualized by two classic markers: γH2AX[49] and 53BP1[50,51] (Fig. 4a). γH2AX forms a specialized chromatin structure that can extend hundreds of kilobases away from the DSB. 53BP1 promotes repair of DSBs by NHEJ as part of the Shieldin complex[51] and suppresses HR. We screened each cell in the tissue slice with NeuN to identify neurons and evaluated whether those neurons also co-stained with γH2AX (green) (Fig. 4a) or 53BP1 (red) (Fig. 4b). NeuN co-stained with γH2AX in all four brain regions, while co-staining in NeuN (−) cells was minimal (Fig. 4a, DAPI). The results indicated that DNA damage in the brain sections was most prominent in neurons (Fig. 4a, b). Although HR proteins were also expressed (Fig. 3), the robust 53BP1 staining (Fig. 4b) in the absence of proliferation strongly suggested that the NHEJ might be a favored DSBR pathway in these cells. Since members of both DSBR pathways (Ku70, Ku80, and MRE11) were expressed in all brain regions (Fig. 3 and Supplementary Fig. 3), it was surprising that DSBs were not completely suppressed in any region of the brain (Fig. 4c). Indeed, a residual population of unrepaired DSBs accumulated with age from 7 to 75 weeks (Fig. 4c) despite expression of DSBR machinery, particularly in the CTX and HIP. To determine whether γH2AX staining reflected actual DSBs, they were directly measured using a neutral comet assay[52,53] in dispersed brain cells from mouse tissue at young (7 weeks) and old (75 weeks) ages (Fig. 4d, e). The DSBs, as measured by comet tail length under non-denaturing conditions, increased with age in all four brain regions (Fig. 4d, e), consistent with the age-dependent increase in γH2AX staining (Fig. 4c). Internal standards[54] confirmed the linear relationship between the comet reactions and the sample (Supplementary Fig. 5a, b). The same results were obtained whether the data were quantified as comet tail length, % comet tail, or comet moment (Supplementary Fig. 5c, d).

The intensity of γH2AX staining in glia was significantly lower than that of neurons and damage levels in these cells were more difficult to measure in tissue slices. Thus, we isolated primary glia from all four brain regions (CBL, STR, CTX, and HIP) and quantified DNA breaks in neutral CometChip assays by measuring the comet tail length (Fig. 5a, examples of digital comet tail images). Although far less frequent than those in neurons (Fig. 4c), DSBs in glia were obvious from the comet tail length (Fig. 5b) and paralleled the γH2AX staining pattern in the same cells (Fig. 5c). The in vitro features of glial cultures appeared to be consistent with their in vivo features: the cultures stained positively with the glial marker, glial fibrillary acidic protein (GFAP) (Supplementary Fig. 6a, b), the DNA repair proteins were expressed prominently in the nucleus (Supplementary Fig. 6a, b), and expression patterns of DNA repair proteins in isolated glial cells and intact tissue glia were similar, as illustrated for MSH6 (Supplementary Fig. 6c, d) and Ku80 (Supplementary Fig. 6e, f). Despite good expression of the DSBR machinery, DSBs in the brain were present in all cell types but were primarily compartmentalized in neurons and accumulated with age.

## DSB formation exceeds DSBR

Given a presumed need to regulate damage, the results raised the key question as to why the residual DSBs escaped repair. Since all brain regions expressed DSBR proteins, it was possible that DSBs accumulated in regions where repair machinery was naturally lowest. However, the opposite was true. In all brain regions, the level of DSBs, as measured by γH2AX staining, was directly proportional to the expression level of Ku80 in the same neurons (Supplementary Fig. 7). DSBs were highest where the level of Ku80 was highest, suggesting that the DSBR machinery was tuned to accommodate the level of damage. Yet, the DSBs were not efficiently removed (Fig. 4c). In the HIP and STR regions, Ku80 expression was similar (Supplementary Fig. 7a). If DSBs observed in the brain tissue depended only on the efficiency of the expressed DSBR machinery, then we expected that DSBs in these two regions would be equivalent. In contrast to the expectation, the number of DSBs was 2–5-fold higher in the HIP relative to the STR (Supplementary Fig. 7b). DSB formation apparently exceeded the DSBR capacity. More than one DNA repair pathway led to a basal population of DSBs in brain cells. Pathways that facilitated DSB formation were as important as those that facilitated DSBR and had to be considered together.

## Base excision contributes to the formation of DSBs in brain cells

DSBR mechanisms have been thoroughly characterized[3,13,17]. However, the pathways contributing to DSB formation in the brain were less clear. The most common source of DSBs occurs during cellular replication, which neurons lack. However, DSBs can arise if SSBs occur simultaneously and in proximity on opposite strands of the DNA double helix. Indeed, transient SSBs form as repair intermediates from multiple pathways (BER, MMR, TCR/NER)[13,17,55]. If SSBs from these pathways led to DSB formation, we expected that pathway contribution would be proportional to pathway activity. We used fluorescent multiplex host cell reactivation (FM-HCR) assays[56–58] to measure the repair activity in primary glial cultures from the four brain regions (Fig. 6a). Briefly, fluorescence reporter plasmids were engineered with DNA lesions that are specifically repaired by one of the major pathways (Fig. 6b). Lesions included a DSB (HR and NHEJ reporters), a tetrahydrofuran abasic site analog (THF; long patch BER), an A:C mismatch (MMR), or ultraviolet (UV) radiation-induced DNA damage (NER) (Fig. 6b). Upon transfection into cells, expression of the reporter gene is restored when the lesions are repaired (Fig. 6a). The efficiency of repair is calculated from fluorescence intensity of the reporter as a percentage of the expression from an equivalent reporter in the absence of damage. The normalization and calculation of repair capacity from flow cytometric data has been described in detail previously[56–58]. A major strength of this approach is the ability to measure the simultaneous activity of multiple DNA repair pathways in parallel in the same cell. We refer to the collective activity of multiple pathways as a "DNA repair landscape".

All DNA repair pathways generating SSB intermediates had activity in the landscapes of cultured glial cells (Fig. 6c). However, the relative activities were striking. BER activity was highest in all regions, and, depending on the region, was roughly 4–10-fold more active than MMR or TCR/NER (Fig. 6c). Thus, BER appeared to be the most significant contributor for SSB level relative to other pathways. Notably, in the same cells, DSBR [NHEJ (~5% reporter expression) and HR activity (~1% reporter expression) were consistently low in all regions of the mouse brain (Fig. 6c). Thus, BER activity was 5–50-fold more active than DSBR in vitro depending on the region. Based on the relative pathway activity, it was feasible that the frequent production of SSBs through efficient base excision promoted the formation of DSBs, but once formed, they were inefficiently repaired and accumulated.

## Oxidative DNA damage promotes the formation of DSBs in the mouse brain

We tested whether SSBs might plausibly lead to DSBs in mouse brains under normal conditions (Fig. 7a). BER had high activity across brain regions (Fig. 6c). The potential for SSB formation from BER was assessed by measuring the abundance of 8-oxo-G, a BER substrate and a common form of endogenous base damage (Fig. 7a). 8-oxo-G

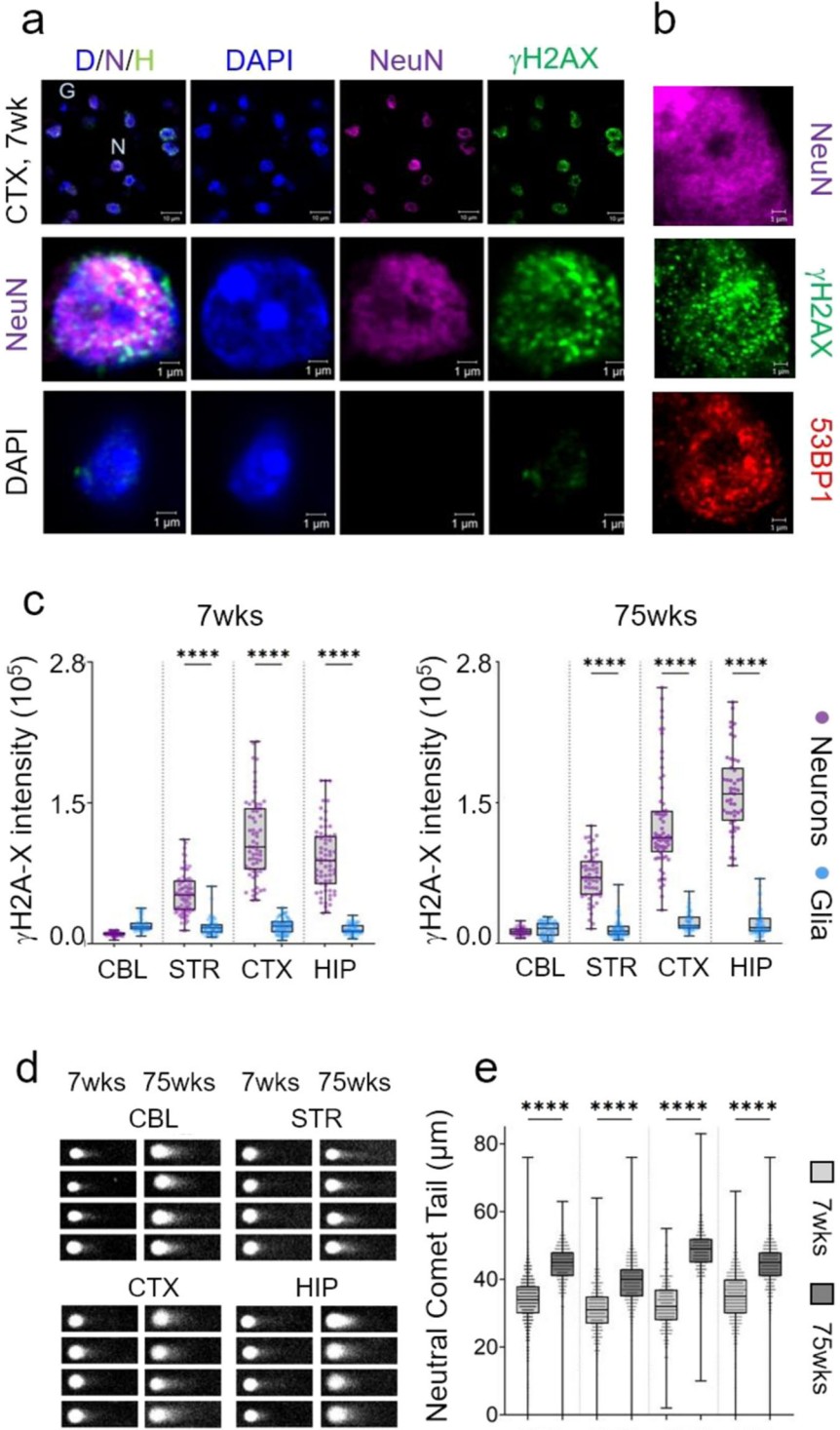

**Fig. 4 | DSBs accumulate in neurons of normal mouse brain. a** Representative IF images of stained neurons and glia in CTX of 7 weeks C57BL/6J male mice. NeuN (purple) and γH2AX (green) prominently co-stain in neurons. Tissue staining in each cell is displayed in a series of separate channel images for DAPI (blue), NeuN (purple), or γH2AX (green), as indicated, or as an overlay of all three (D/N/H) (left). (top row) Field of stained cells for DAPI, NeuN, and γH2A-X antibodies (D/N/H) (Scale bar is 10 μm); neurons co-stained with NeuN and DAPI, while glia stained only with DAPI. Magnified images of Neuron (N) (middle row) and glia (G) (bottom row) were selected from the cell fields in the top row (Scale bar is 1 μm). **b** The DSB markers γH2AX (green) and 53BP1 (red) were identified as neuronal if they co-stained with NeuN (purple) in male C57BL/6J mouse tissue; γH2AX (green) and 53BP1 (red) markers in glia co-stained only with DAPI (blue). **c** Quantification of IF staining intensity for γH2AX from *n* = 50 randomly selected cells of each type in *n* = 3 tissue sections of 7 (left) or 75 (right) weeks animals (from **a**), respectively. Data are displayed as a box and whisker plot, where the box are 25–75% of the values, the line indicates the median value, and 25% maximum values and 25% minimum values are indicated by whiskers above and below the box, respectively. The probability statistics for comparing significance among regions were determined from a one-way ANOVA are ****$P < 0.0001$ for all type comparisons. **d** Neutral CometChip assay results for dispersed cells from different brain tissues (CBL, STR, CTX, HIP) at 7 weeks and 75 weeks. **e** Quantification of the comet tail length (μm) in dispersed cells from (**d**) as a function of age; 7 weeks, gray; 75 weeks, black. Points shown are individual scored comets from at least *n* = 550–890 cells per tissue section (*n* = 3 animals). The probability statistics for comparing the Neutral CometChip Tail lengths of dispersed cells at the two ages were determined from a one-way ANOVA. ****$P < 0.0001$ for all comparisons.

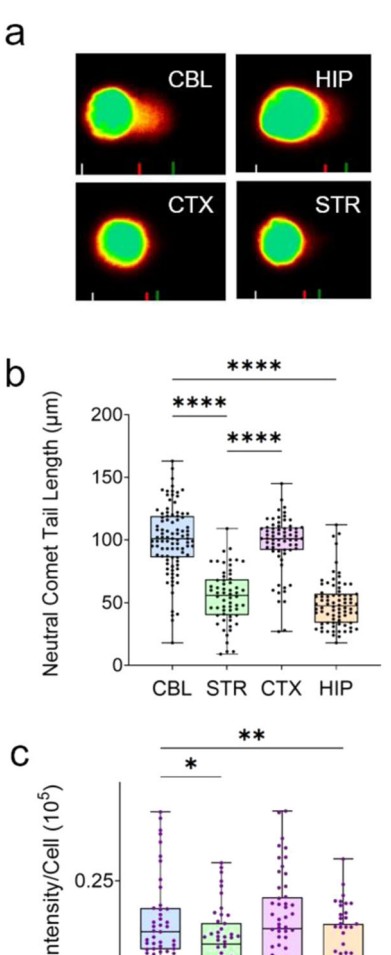

**Fig. 5 | Neutral CometChip assay in cultured glial cells from CBL, HIP, CTX, and STR. a** Representative digital images of comet tails (orange) (μm) from a neutral CometChip assay in cultured glial cells from CBL, HIP, CTX, and STR. The software delineates the features of each identified comet, as indicated by the lines at the bottom of each image. The region bounded by the white and red lines signifies the head portion of the comet, while the space between the red and the green lines illustrates the tail section. The green line extends to the fluorescence intensity distribution of the tail region, eliminating the background intensity. **b** Result of comet tail length (μm) for cultured glial cells from CBL (blue), HIP (orange), CTX (pink), and STR (green) of dissected tissue of $n = 4$ animals. Points are individual scored comets. At least $n = 50$ cells per regional glial line were scored in three technical replicates ($n = 3$) measured on separate days. Data are displayed as a box and whisker plot, where the box are 25–75% of the values, and 25% maximum values and 25% minimum values are indicated by whiskers above and below the box, respectively. The line in the box indicates the median value. Regional comparions were determined using a one-way ANOVA. ****$P < 0.00001$ for all comparisons. **c** Parallel γH2AX staining intensity in duplicate plates of cultured glial cells from (**b**); CBL (blue), HIP (orange), CTX (pink), and STR (green) displayed as a box and whisker plot as in (**b**). Regional comparions were determined using a one-way ANOVA, *$P = < 0.01$, **$P < 0.001$.

nucleotides in tissues were enzymatically released from genomic DNA and their level was detected with a fluorescence-based antibody assay (Fig. 7b)[54,59]. The level of 8-oxo-G/μgDNA was robust between 7 and 75 weeks indicating that each brain region had a constitutive pool of oxidized BER substrates as a source for SSBs (Fig. 7c, d). Indeed, SSBs are common in the genome. The alkaline CometChip assay under denaturing conditions confirmed that SSBs were present in each

region at young (7 weeks) and old (75 weeks) ages (Fig. 7e, f) and increased with age concomitantly with DSBs that were detected by neutral comet (Fig. 4d, e).

DSBs would not be expected from excision of oxidized bases unless they arose from two closely opposed SSBs (Fig. 8a). To test whether SSB to DSB conversion was possible in vivo, we lowered the level of oxidized bases by treating mice with the mitochondrially directed antioxidant, XJB-5-131, and determined whether ROS suppression had an impact on DSB formation (Fig. 8a). XJB-5-131 is a bifunctional molecule comprising a peptide delivery component (Fig. 8b, black and black hatched box)[60-63], which directly targets the mitochondrial membrane and delivers an antioxidant nitroxide, 2,2,6,6-tetramethyl piperidine-1-oxyl (TEMPO) (Fig. 8b, red and red hatched box) to neutralize intracellular ROS. In this experiment, mice were aged to 60 weeks to allow significant formation and accumulation of basal DSBs, after which, the animals were treated with XJB-5-131 for another 30 weeks. Indeed, in all four regions of the brain, XJB-5-131 treatment suppressed γH2AX staining by 30–50% relative to neurons of vehicle-treated mice (Fig. 8c). The results provided definitive evidence that a significant fraction of DSBs depended on DNA oxidation, supporting the idea that excision of oxidized bases promoted SSBs which were then converted to DSBs in vivo (Fig. 8a, c).

### Oxidative DNA damage promotes SSB to DSB conversion

As further support for the SSB to DSB conversion mechanism, we performed three additional in vitro experiments as proof-of-principle. Since adult neurons are difficult to culture or to transfect, mechanistic tests in mouse embryonic fibroblast cells (NIH3T3) were designed to report on features that would also occur in neurons. In the first experiment, we repeated the animal experiment in cells (Fig. 9a). Confluent cultures of immortalized NIH3T3 cells were treated with peroxide to elevate ROS in the presence or absence of the ROS inhibitor, XJB-5-131 (Fig. 9a). If SSB to DSB conversions depended on the oxidative stress, as we observed for animal brains in vivo, we expected that peroxide treatment would elevate DSBs, and that the rise would be suppressed in the presence of XJB-5-131. Although peroxide treatment most often leads to oxidation of single bases, the transfected cells stained robustly with γH2AX and 53BP1 foci within 30 min of treatment (Fig. 9b, c −XJB-5-131). DSB foci were suppressed in the presence of XJB-5-131 (Fig. 9b, c; +XJB-5-131, black, −XJB-5-131, gray). No γH2AX and 53BP1 marker staining was observed in the absence of peroxide treatment (Supplementary Fig. 8a). Thus, DSB formation depended on base oxidation and appeared to arise from conversion of two SSBs that formed after BER removal of the damage in vivo and in vitro.

In a second experiment, we tested whether the reduction in DSBs by XJB-5-131 could occur in the absence of base oxidation (Fig. 9d–f). The premise being that SSB to DSB conversions would be overestimated if non-oxidative lesions contributed to the population of DSBs. To test this idea, cell cultures were prepared as in experiment 1 but, in this case, the SSBs were generated enzymatically by transient transfection of recombinant Cas9D10A[64]. This mutation converts the clustered regularly interspaced short palindromic repeats (CRISPR) Cas9 nuclease into a "nickase" capable of introducing only one SSB per reaction[64] in genomic DNA of cells. We targeted SSB formation to the mouse Major satellite (MajSat) using guide RNA that was complementary to the site (Fig. 9d–f). MajSat DNA is an integral part of heterochromatin (see Fig. 9j) and is obvious in the bright DAPI-staining nuclear domains that co-stain with repressor histone modifications H3K9Me3 (see Fig. 9k). In the transfected cells, we tested whether (1) SSBs could be converted to a DSBs in the absence of exogenously introduced ROS and (2) whether DSB formation under these conditions could be suppressed by XJB-5-131 (Fig. 9d–f). Although Cas9D10A induces only SSBs, the transfected cells stained robustly with γH2AX

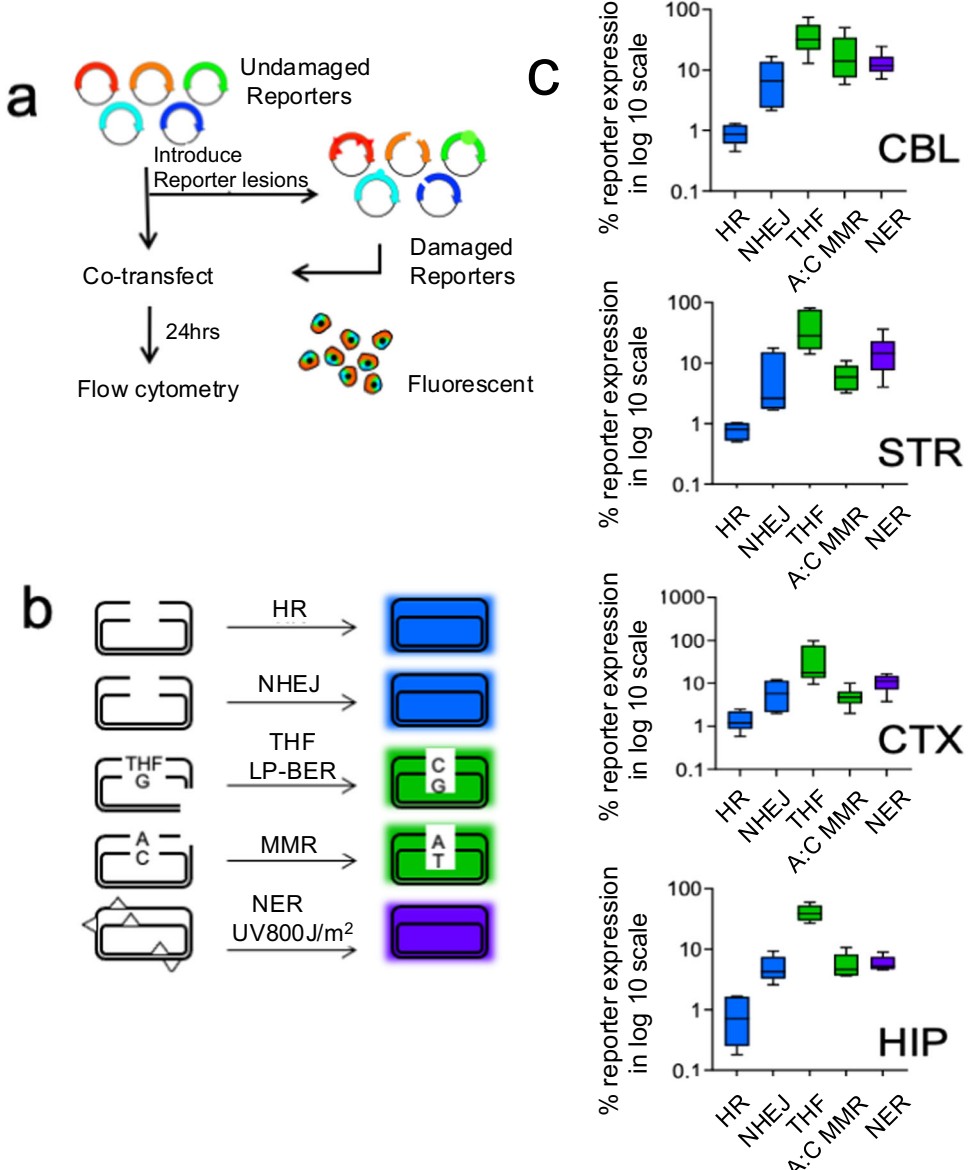

**Fig. 6 | Primary brain cells have high BER and low DSBR in the mouse brain.**
**a** Fluorescence multiplex host cell reactivation (FM-HCR) assay was used to determine the activities of different DNA repair pathways in glial cultures from the CBL, STR, CTX, and HIP. The indicated DNA repair pathways are described in the text. Primary glia was seeded and transfected with sufficient efficiency (4–8%) to afford a robust analysis of fluorescence intensity by flow cytometry[56–58] and DNA repair capacity in the transiently transfected cells. **b** Color coded reporter plasmids for the indicated pathways. **c** Reporter activity plotted as a % reporter expression of repair pathway for four brain regions, as indicated. Repair activities were measured in plated glial cultures; $n = 6$ for CBL and STR, $n = 5$ for CTX, and $n = 4$ for HIP (except CBL HR, $n = 5$; STR HR, $n = 4$; CTX NER and HR, $n = 4$). Source data are provided in Source data file. Data are displayed as a box and whisker plot, where a box indicating the 25th to 75th percentile values, the line is the median, with the lower whiskers representing the minimum 25% of data values and the whiskers above the box representing the 25% maximum values.

and 53BP1 markers within 6 h post transfection (Fig. 9e, f, −XJB-5-131). The results confirmed that SSB to DSB conversions had occurred. Furthermore, the IF signals of γH2AX and 53BP1 co-localized at heterochromatin regions indicating that SSBs and DSBs formed at the same target sites (Fig. 9e). Indeed, γH2AX and 53BP1 marker staining was present in DAPI-staining foci at domains consistent with the H3K9Me3 heterochromatin marker (Fig. 9e, H3K9Me3), as expected for MajSat repeats (Fig. 9j). Notably, the formation of Cas9D10-directed DSBs was not suppressed by treatment of cells with XJB-5-131 (Fig. 9e, ±XJB-5-131) (Fig. 9f; +XJB-5-131, black, −XJB-5-131, gray). No γH2AX and 53BP1 marker staining was observed in the absence of Cas9D10A transfection (Supplementary Fig. 8b). Collectively, the results from experiments 1 and 2 demonstrated that DSB formation

arose from base oxidation and was suppressed by XJB-5-131, supporting a SSB to DSB conversion mechanism.

In a third experiment, we tested whether XJB-5-131 had any impact on DSB formation itself (Fig. 9g–i). If XJB-5-131 inhibited DSB formation directly, for example, then its reduction might be unrelated to DNA base oxidation. We repeated experiment 2 but replaced Cas9D10A with active Cas9 and targeted it to the MajSat DNA using Cas9 guide RNA. The Cas9 nuclease cuts duplex DNA without an SSB intermediate or DNA oxidation (Fig. 9g). Thus, we expected that DSBs would form under these conditions independently of XJB-5-131 treatment (Fig. 9g). Indeed, DSBs formed in these cells within 6 h post transfection, as detected by IF intensity of γH2AX and 53BP1 staining (Fig. 9h, −XJB-5-131). No γH2AX and 53BP1 marker staining was observed in the absence

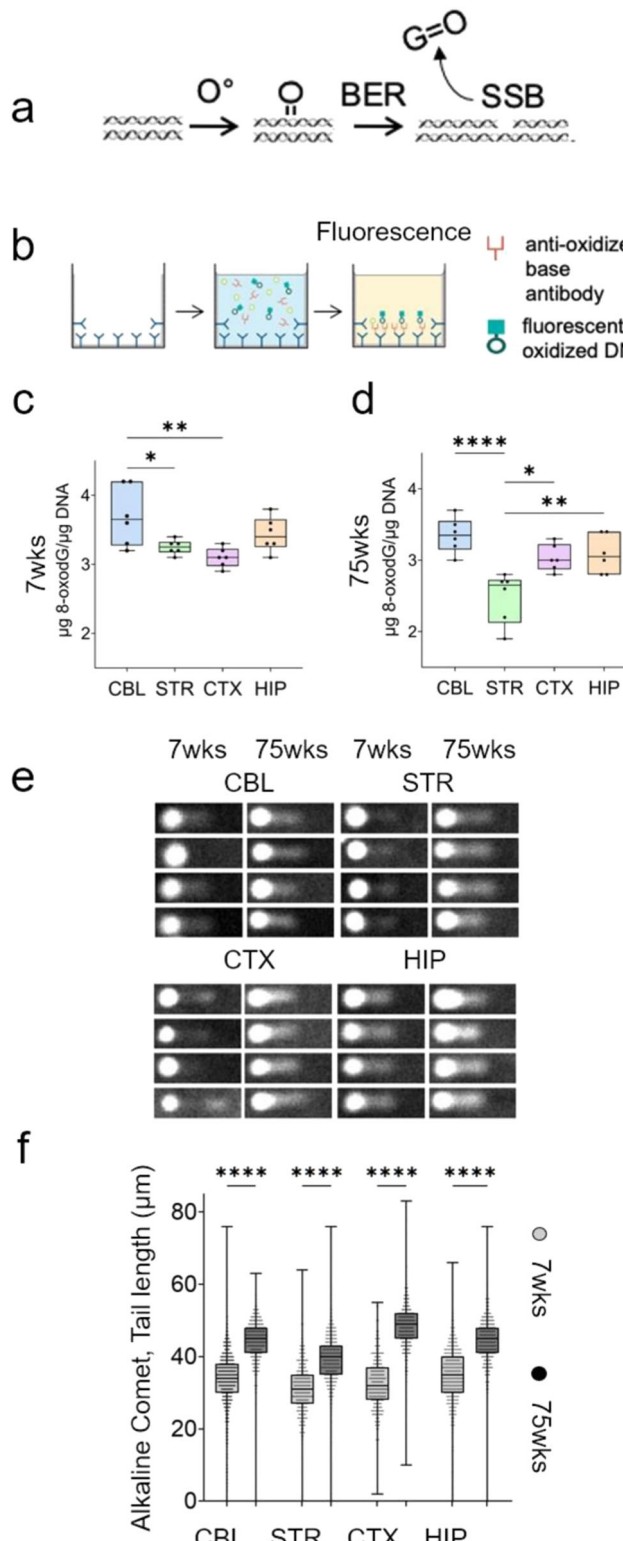

**Fig. 7 | DSBs depend on base excision of oxidized bases in vivo. a** Schematic illustration of a transient SSB intermediate generated by excision of oxidized base damage during BER. **b** Schematic diagram of the 8-oxo-G ELISA assay steps. **c**, **d** Quantification of 8-oxo-G in cells from mouse brain (*n* = 3) in four regions, CBL (blue), STR (green), CTX (pink), and HIP (orange) at 7 (**c**) and 75 (**d**) weeks using the competitive ELISA assay (Cayman Chemical). Data are displayed as a box and whisker plot, where the line indicates the median value, the box are 25–75% of the values, and 25% maximum values and 25% minimum values are indicated by whiskers above and below the box, respectively. The probability statistics for regional comparisons was determined from a one-way ANOVA, *$P < 0.01$, **$P < 0.001$, ***$P < 0.0001$. **e** *n* = 4 technical replicates of Alkaline CometChip images illustrate the presence of SSBs in dispersed cells from dissected brain regions (CBL, STR, CTX, HIP) of 7 or 75 weeks animals. **f** Analysis of the comet tail length for (**e**) at 7 weeks (gray) and 75 weeks (black). Each symbol represents an individual scored comet (*n* > 700). SSBs increase with age in all brain regions. Data are displayed as a box and whisker plot as defined in (**c**, **d**). Source data are provided in Source data file. The probability statistics for age comparisons in each region was determined from a one-way ANOVA. ****$P < 0.00001$.

results of the three complementary experiments provided evidence that excision of oxidized bases promoted reversible SSB to DSB conversion in vitro and in vivo.

## Discussion

Under normal metabolic conditions in vivo, it is estimated that 70,000 DNA lesions occur in genomic DNA per cell per day[55,65–67]. Most of these are oxidation-induced SSBs[55,65–67], which are constitutively generated throughout life (Figs. 7 and 8). Thus, in vivo conditions provide ample opportunity for SSBs to convert into DSBs and accumulate over a lifetime. However, if left unchecked, DSB accumulation would be unrestricted, and neurons would be at risk for toxicity (Fig. 10a)[67]. By integrating repair protein expression, DNA repair activity, and lesion level in the brain, we find that the brain expresses the same repair machinery as in other cell types, and repair of SSB and DSBs uses canonical mechanisms. However, the number of DSBs is not solely determined by DSBR activity. Rather, in the basal state, repair activity is unequal and is characterized by high BER and low DSBR. This natural imbalance creates overarching conditions whereby, the high level of closely spaced SSBs on opposite side of the DNA helix can passively convert to DSBs (Fig. 10a). If SSBs occur too frequently[66] or DSBs persist too long[67,68], the equilibrium is likely to shift far to the right, leading to aging or disease (Fig. 10a). In both animals and in cells, increases in oxidation push the equilibrium towards DSBs, which are inhibited by suppression of ROS (Figs. 8 and 9). We propose that neurons coordinate DNA repair and utilize SSB-DSB conversion as a check point mechanism to modulate the overall formation of DSBs in the absence of replication. Strong BER removal of oxidized bases coupled with inefficient DSBR maximize SSB-DSB conversion and ensure that residual unrepaired DSBs are generated in sufficient numbers for possible functional uses (e.g., transcriptional signaling)[4,41–45]. At the same time, reduction in the oxidation state cooperates with low DSBR to keep DSBs from exceeding normal limits (Fig. 10a).

Emerging evidence suggests that such a mechanism is plausible. SSB-DSB conversion arises when two SSBs occur in proximity on opposing sides of the DNA helix. Indeed, recent sequencing technologies confirm that SSBs are not random[68–72]. Rather, they are closely clustered in the regulatory regions of mammalian genes. Repair-Seq[70] and Synthesis-associated with repair sequencing (SAR-Seq)[71] utilize incorporation of the thymidine analog EdU to provide a biotinylated capture tag for DNA damage. These tags notably led to predominant capture of SSB repair machinery in post-mitotic neurons[70,71]. SAR-Seq peaks co-localized with Poly (ADP-ribose) polymerase (PARP) and X-Ray Repair Cross Complementing 1 (XRCC1), which is a protein scaffold required for BER and SSB repair[70]. Co-localization indicates

of transfection (Supplementary Fig. 8b). Notably, XJB-5-131 treatment did not have an impact on DSB formation (Fig. 9h, +XJB-5-131); (Fig. 9i, +XJB-5-131, black, −XJB-5-131, gray). Independent of treatment, DSBs were present in foci (Fig. 9h) in domains that co-stained with DAPI and heterochromatin markers (9j, k, H3K9Me3). Since DSBs were enzymatically produced by Cas9 or Cas9D10A, the production of DSBs was independent of DNA repair protein expression or repair activity in the cell. The results suggested that DSBs in both cell types arose by conversion of SSBs and likely by the same mechanism. Collectively, the

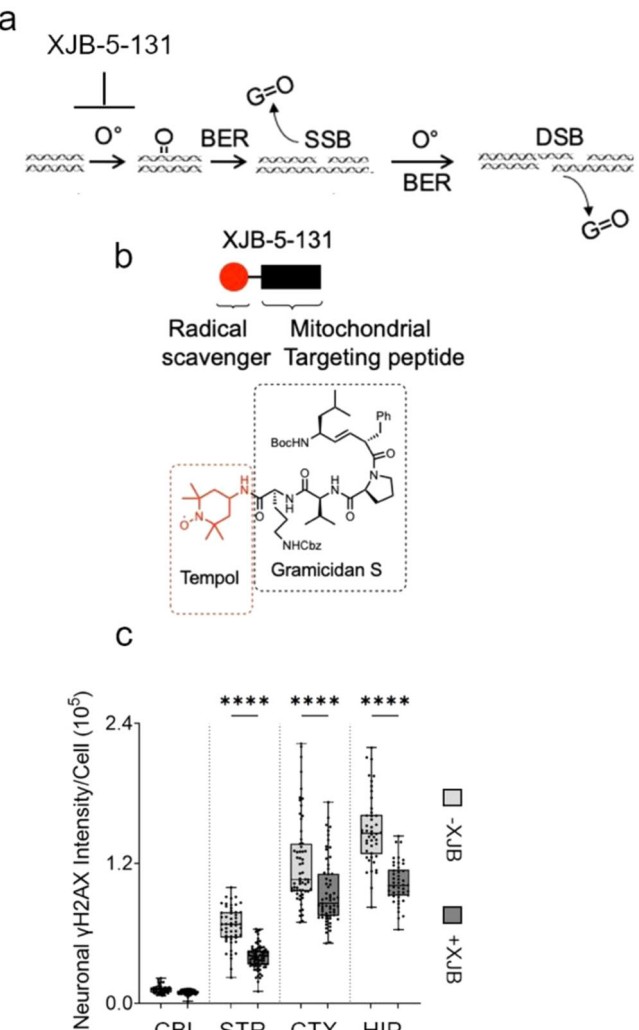

**Fig. 8 | DSBs depend on base excision of oxidized bases in vivo. a** Schematic diagram of DSB formation from two "in trans" SSBs leading to DSBs after excision of oxidized bases during BER. **b** Diagram of functional parts of the XJB-5-31 ROS inhibitor and its chemical structure. The red ball represents the tempol antioxidant (red) (red hatched box), which is fused to a mitochondria-targeted carrier peptide (black) (black hatched box)[60–63]. The target carrier peptide is based on the Gramicidin S, an antibiotic that targets the mitochondrial membrane directly. **c** Single-cell quantification of DSBs in brain tissue sections from vehicle (gray) ($n = 3$) or XJB-5-131 treated (black) ($n = 3$) male *C57BL/6J* mice as measured by IF signal intensity of cells that co-stain with NeuN and γH2AX. In each tissue section, the DSB marker γH2AX was quantified in 50 randomly selected NeuN positive neurons from CBL, STR, CTX, and HIP of ($n = 3$) treated male *C57BL/6J* mice of 80–90 weeks (black) and $n = 3$ vehicle treated control mice of comparable age (gray). The regional-specific comparisons of DSB marker level were evaluated using a one-way ANOVA. Treated mice were characterized by a statistically significant reduction in γH2AX signal intensity relative to vehicle treated animals, ****$P < 0.00001$.

that SAR-Seq peaks contain SSBs or at least serve as a repository of the SSBR machinery, most often from BER[70,71]. Most striking, however, is that SSBs accumulate at unexpectedly high levels in close proximity at specific sites within enhancers, or near CpG dinucleotides and sites of DNA demethylation[70,71]. Thus, clustered SSBs are poised for conversion to DSBs in transcriptional regulatory sites in gene promoters (Fig. 10b)[70–76].

In other systems, DSB-bound complexes in enhancers stimulate transcriptional programs to reversibly modulate biological transactions[4,41–46,73–77]. For example, topoisomerase II (TOP2)-mediated DSBs can stimulate early response genes (c-*Fos*, *FosB*, *Npas4*, and *Egr*1)[3,4,73] which serve as functional adapters for transient modulation of beneficial downstream activity[53,73–77]. Indeed, cells lacking *c-Fos* have difficulties in DNA repair[78]. DSBs are used as a regulatory element across organisms. For example, a copper-responsive two-component system named DsbRS has been identified in the important human pathogen *Pseudomonas aeruginosa*[74]. In this system, DSBR acts as a sensor histidine kinase, and DsbR, its cognate response regulator, directly induces the transcription of genes involved in protein disulfide bond formation (i.e., the *dsbDEG* operon and *dsbB*). A similar regulatory strategy appears to operate in mammalian neurons. The DSB-associated factors NPAS4-NuA4[4] initiate a DNA repair pathway for feedback prevention of breaks in activated mammalian neurons[4]. In these documented examples, the DSB-bound complexes act as sensors[4] in that they recognized DSBs and initiate beneficial transcriptional signals for downstream regulation. Whether a transcriptional signal is used to control the SSB-DSB conversions in response to oxidative DNA damage remains to be confirmed. However, a DSB-stimulated transcriptional signal that reduces oxidative damage would aid in shifting the SSB-DSB equilibrium to the left, ensuring that DSBs did not exceed an acceptable threshold for healthy biological transactions (Fig. 10b).

There are many considerations in establishing whether oxidation-induced SSB-DSB equilibria serve as check points. In cells, there are other sources of DSBs that do not require base oxidation. However, known mechanisms are unlikely to significantly influence the SSB-DSB equilibrium in neurons under basal conditions. For example, SSB intermediates from MMR, BER, and NER can be converted to DSBs if they collide with replication forks, but neither fork collapse nor replication-transcription collision occurs in neurons[79]. NER and TCR play roles in repair of 'unscheduled' R-loops in disease cells, which can produce DSBs in replicating cells[80]. However, mounting evidence indicates that normal cells harness the presence of RNA–DNA hybrids in scheduled, 'regulatory' R-loops and promote DNA transactions, including transcription[81–84]. Here, we show that XJB-5-131 reduces DSBs only if they arise from oxidative damage (Figs. 8 and 9d–f). DSBs can arise directly from oxidative damage, but at low frequency compared to BER-induced SSBs. Although NER is also capable of excising oxidized bases, this pathway generates a riskier single-strand gap, and is primarily harnessed to remove helix-distorting oxidative lesions. Collectively, our data support the idea that BER is the most common source of SSBs and that conversion to DSBs is modulated by oxidative state. Experimentally, the SSB and DSB measurements are robust and are validated by γH2AX staining and comet assay (Figs. 4 and 7). While we have measured only 1–2 key components to assess pathway protein expression, repair pathway activity was determined independently in cells within the context of all assembled pathway machinery. All repair activity is measured together within 24 h in an FM-HCR assay (Fig. 6)[56–58]. Although we cannot exclude that kinetics of lesion repair in each reporter plasmid contributes to some variability in activity, the FM-HCR reporters accurately reflect repair activity in other well-characterized cell lines using the same methods with reliable results[56–58].

How neurons keep DSBs controlled over a lifetime has been poorly understood. Replication check points in dividing cells serve as guardians of the genome by regulating repair of DSBs encountered at the fork. In its absence, however, we envision that non-dividing cells are tuned to control the overall formation of DSBs. This type of mechanisms takes advantage of metabolic flux, consistent with the fact that DSBs are elevated in animals during exercise when ROS is high and revert to baseline when animals are at rest[42,43], and with our finding that DSBs constitute a balance, whereby pathways that generate DSBs are as important as DSBR. Whatever the detailed mechanism, modulation of DNA breaks in brain cells appears to critically depend on communication of DNA repair and metabolism. Evidence for direct connections between these two processes is only emerging, but molecular

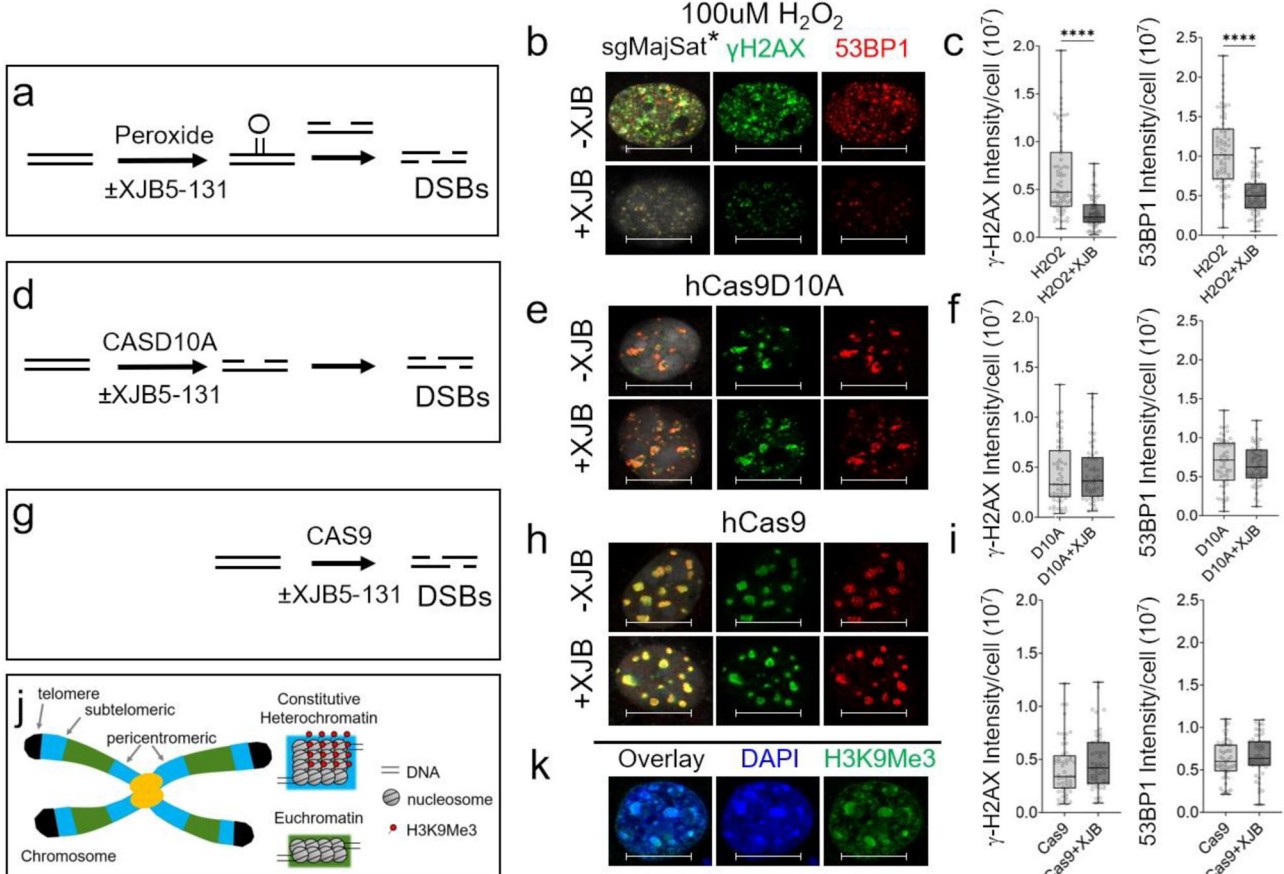

**Fig. 9 | Excision of oxidized bases induces SSB to DSB conversion in vitro.** Three experimental tests for SSB to DSB conversion. More than $n = 50$ NIH3T3 were identified as neurons by NeuN staining and positive cells were subsequently imaged for γH2AX or for 53BP1 in each treatment regime for $n = 3$ replica platings. Scale bars were 10 μm for all images. **a** Schematic design for experiment 1 as described in text. NIH3T3 cells were transfected with "sense" major satellite guide RNA (sgMajSat) and treated with peroxide for 30 min with (+XJB-5-131, 200 μm) or without XJB-5-131 (−XJB-5-131). The sgMajSat* transfection was included in experiment 1 only to ensure that DSB quantification in (**b**, **e**, **h**) occurred under the same conditions. **b** (Top panel −XJB-5-131); IF results for DSB formation in selected cells as detected by staining intensity with the γH2AX (green, middle panel) and 53BP1 (red, right panel) DSB markers antibodies. An overlay of the two (yellow foci, left panel) in the absence of XJB-5-131. (Bottom, +XJB-5-131). Same as (top) but in the presence of XJB-5-131. The low blue emission from mBFP expressed from the sgMajSat* RNA expression vector is represented as gray so as not to be confused with DAPI (blue). **c** Quantification of (**b**) in $n = 50$ cells; gray (−XJB-5-131) and black (−XJB-5-131) in $n = 3$ replicate platings. **d** Schematic design for experiment 2 as described in text. XJB-5-

131 inhibition of DSBs requires base oxidation. **e** IF result for DSBs after transfection of the CASD10A nickase targeted to major satellite DNA using sgMajSat RNA guide ± XJB-5-131, as indicated. The IF signals of γH2A-X (green, middle panel) and 53BP1 (red, right panel) target and co-localized at major satellite loci in heterochromatin domains[88] (see **j**) within 6 h post transfection ± XJB-5-131. **f** Same as (**c**) but for Quantification of $n = 50$ cells from (**e**). **g** Schematic design for experiment 3 as described in text. **h** Same as (**c**) but for CAS9. **i** Quantification of 50 cells from (**i**) is as described in (**f**). Data are expressed as median (line), the box indicating the 25th and 75th percentile with whiskers indicating the minimum and maximum values. The probability statistics in (**c**), (**f**), and (**l**) were derived from a one-way ANOVA, ****$P < 0.0001$ for all cell type comparisons in (**c**, **f**, **i**). **j** Schematic image of heterochromatin domains. **k** IF detection of heterochromatin domains detected by the histone H3K9Me3 antibody (right panel), DAPI (middle panel), and an overlay of both (left panel). NIH3T3 cells have heterochromatin domains and stained positively with the histone H3K9Me3 antibody. Shown is a representative example. This was checked in >10 cells from each of 3 separate cell cultures, Scale bars represent 10 μm.

mechanisms will be important to elucidate and are likely to be broadly relevant.

## Methods

### Animals

Our research complies with relevant ethical regulations. The Institutional Animal Care and Use Committee approved all procedures. Animals were treated under guidelines for the ethical treatment of animals and approved by IACUC protocol #274005 at Lawrence Berkeley Laboratory. All animal work was conducted according to national and international guidelines. Male C57BL/6J were used in every experiment and referred to as C57BL/6J or wild-type (WT) mice (Jackson Labs cat#000664). The Male C57BL/6J were used as controls in an ongoing study using a disease model where males were used to keep gender consistency. At least 3 animals ($n = 3$) were used in each tissue experiment.

### Brain tissue lysate preparation

Dissected brain tissues were flash-frozen in liquid nitrogen and were preserved at −80 °C until lysate preparation. Brain tissue was placed in a microcentrifuge tube and thawed on ice for ~10 min before adding lysis buffer consisting of T-PER Tissue Protein Extraction Reagent (Thermo Scientific, Cat No. 78510), ~8.7x Halt protease inhibitor cocktail (from 100x, Thermo Scientific, Cat No. 78430), ~4.3x Halt phosphatase inhibitor cocktail (from 100x, Thermo Scientific, Cat No. 78420). After adding the lysis buffer, the tissue sat on ice for 2–3 min before being triturated with a Bel-Art extended handle pestle (Millipore-Sigma, Cat No. BAF199210001). The triturated tissue sample was centrifuged at $16,100 \times g$, 4 °C for 1.5 min in Eppendorf 5425R refrigerated microcentrifuge. Following the centrifugation, the sample was subject to 4 cycles of pulse sonication on ice, performed in which 2 cycles of 10 s-on and 30 s-off followed by centrifugation at $16,100 \times g$, 4 °C for 1.5 min, 1 cycle of 10 s-on and 30 s-off followed by

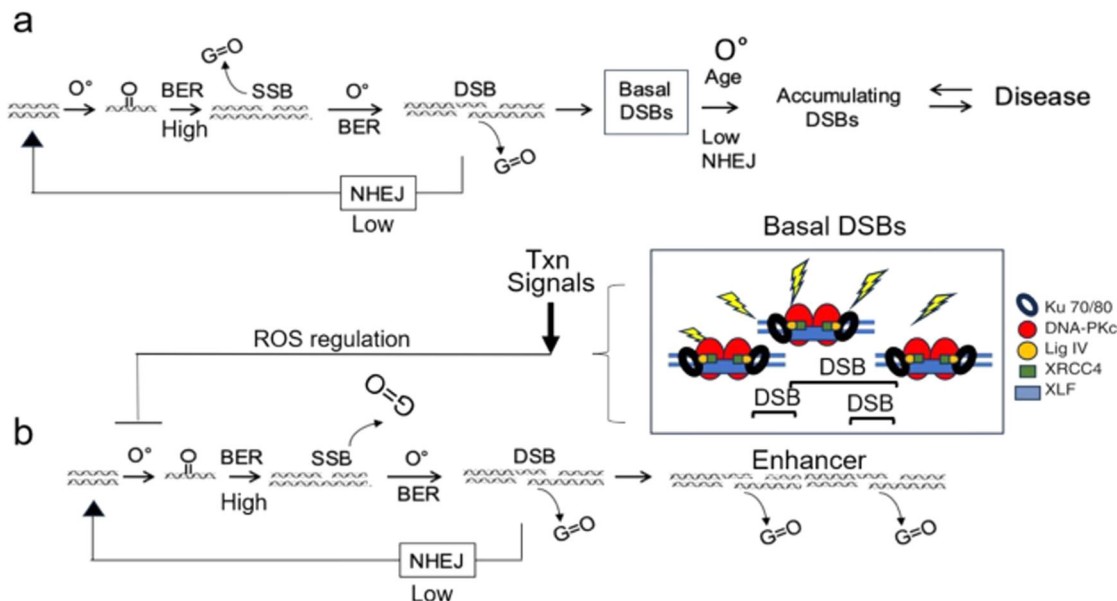

**Fig. 10 | A passive SSB to DSB equilibrium model for damage check points in non-dividing cells. a** Equilibrium model for SSB to DSB conversion. **a** Schematic diagram of the equilibrium model for modulating DSB formation in neurons, as described in text. Switching between SSB and DSB states is modulated by oxidation. BER activity of oxidized bases is high and generates substantial SSBs, which, if closely spaced on opposite strands, can be passively converted to DSBs. Decreases in oxidation reduce BER-derived SSBs and suppress unrestricted DSB accumulation. If SSBs occur too frequently or DSBs persist too long, the equilibrium shifts to the right, leading to aging or disease. **b** Model for reduction of oxidation-induced DSBs in enhancers. DSBR is low in neurons, and reversing the equilibrium to lower the DSB level may require additional support. As observed in other systems, the DSB-bound machinery may initiate signals to stimulate transcriptional programs and reduce ROS, thereby shifting the SSB-DSB equilibrium to the left.

centrifugation at $16,100 \times g$, 4 °C for 1.5 min, and then 1 cycle of 10 s-on and 30 s-off. Sonication was performed using Branson Sonifier Cell Disruptor 185. The resulting homogenate was centrifuged once more at $16,100 \times g$, 4 °C for 30 min and supernatant was used for the protein quantification assay, SDS-PAGE and western blotting analyses.

## Protein quantification assay

The protein concentration of each lysate was quantified using Pierce 660 nm Protein Assay reagent (Thermo Scientific, Cat No. 22660). Briefly, a small volume of each lysate was diluted 1:10 in phosphate-buffered saline (PBS). 10 μL of diluted lysate was mixed with 340 μL of Pierce 660 nm Protein Assay reagent in a 1.7 mL microcentrifuge tube and then incubated at room temperature (RT) for ~5 min before loading 150 μL of each mixture in each of 2 wells of a flat bottom, transparent 96-well plate. For each lysate, a mixture was prepared and analyzed in triplicate by Infinite M1000 microplate reader (Tecan) using Tecan i-control software.

## SDS-PAGE and western blot

SDS-PAGE samples were prepared by mixing brain tissue lysate containing 60 μg total protein (20 μg for apurinic/apyrimidinic endonuclease 1 or APE1, and 10 μg for glyceraldehyde 3-phosphate dehydrogenase or GAPDH), T-PER Tissue Protein Extraction Reagent (to level the volume of all the samples), 1x NuPAGE LDS Sample Buffer (from 4x, Invitrogen, Cat No. NP0007) and 1x NuPAGE Sample Reducing Agent (from 10x, Invitrogen, Cat No. NP0009). Each sample was boiled at ~95 °C for 10 min and centrifuged at $15,000 \times g$, at RT for 5 min. Total proteins in each SDS-PAGE sample were subsequently resolved along a Novex WedgeWell Tris-Glycine 4–12% Mini gel (Thermo Fisher, Cat No. XP04205BOX) in XCell SureLock Mini-cell Electrophoresis system (Thermo Fisher). The resolved proteins were transferred to a nitrocellulose membrane using Trans-Blot Turbo Transfer System (Bio-Rad), with a standard protocol (25 V, 1.0 A, 30 min). The resulting nitrocellulose membrane was blocked in 5% non-fat dry milk in PBS + 0.05 % Tween-20 (PBST) for 1 h at RT. The membrane was then incubated in 5% non-fat dry milk in PBST + primary antibody for 1 h at RT. The membrane was then washed 3 times with PBST. The membrane was subsequently incubated in 5% non-fat dry milk in PBST + secondary antibody for 30 min at RT. The membrane was next washed 3 times with PBST and once with PBS. Lastly, Amersham ECL Select Western Blotting Reagent (Sigma-Aldrich, Cat No. GERPN2235) was applied to the membrane and the western blot image was developed using VersaDoc MP 4000 Imaging System (Bio-Rad) and Quantity One 1-D Analysis software (Bio-Rad). Quantification of protein band intensity was performed using Image Lab software (Bio-Rad). The specific antibodies used in the analysis are listed in key resources Supplementary Table 2 and antibody testing is provided in Supplementary Table 3.

## Brain tissue sections

We collected brains from male mice (7–10 weeks and 70–90 weeks of age) of *C57BL6/J* mice. Brains were cut into 4 coronal sections and arranged in a holder filled with OCT (Tissue-Tek O.C.T. from Sakura), and immediately frozen in isopentane bath cooled by liquid nitrogen, prior to storage at −80 °C. This arrangement of the tissue permitted concurrent cutting of all 4 sections at a time. Cuts were made so that all the relevant regions (caudoputamen of striatum, CA1 region of hippocampus, the granular and molecular layer of Crus1 of the cerebellum, and the entorhinal area of the cortex) were present in each cut. Sectioning onto slides (Histobond from VWR) was performed on a cryostat (Leica CM1950) using cut settings (chuck = −14 °C, blade = −15 °C) and cutting 10–15-μm-thick sections. Slides were air dried (15 min) and stored at −80 °C until use. In parallel, the dissected tissue of equal weight before freezing was dispersed for cell number counting on a hemocytometer.

## Immunofluorescence staining and imaging

Primary antibodies used were mouse anti-NeuN Alexa-488 conjugate (Millipore #MAB377X) (1:500), mouse anti-GFAP Cy3 conjugate (Abcam #ab49874) (1:500), mouse anti-APE1 (Novus #13B8E5C2)

(1:500), mouse anti-Ku80 (Santa Cruz #515736) (1:500), mouse anti-ERCC1 (Santa Cruz #17809) (1:500), rabbit anti-MSH2 (Abcam #92473) (1:500), mouse anti-MSH3 (Millipore #MABE324) (1:500), rabbit anti-MSH6 (Abcam #ab92471) (1:500), rabbit anti-XPA (AbClonal #A1626) (1:500) and rabbit anti-MRE11 (Novus #NB100-142) (1:500) (Table S1). Secondary antibodies used were donkey anti-mouse Alexa-488 (Jackson #715-545-150), goat anti-mouse Alexa-568 (Invitrogen #A21124), donkey anti-rabbit Alexa-488 (Jackson #711-545-152), and goat anti-rabbit Alexa-555 (Invitrogen #A32732). Anti-mouse antibodies were tested and selected for those having the least amount of background staining, which was typically visible as staining of blood vessels amongst various commercially available options.

Brain sections on slides were thawed and fixed with 4% PFA for 20 min at 4 °C, then washed once with PBS. They were then pre-extracted with RNase in CSK buffer, i.e., 0.3 mg/mL RNase A (New England Biolabs) in (10 mM PIPES pH 7.0, 100 mM NaCl, 300 mM sucrose, 3 mM MgCl$_2$, 0.7% Triton X-100). Lipofuscin autofluorescence was quenched by soaking in 1x TrueBlack (CellSignal #92401) in 70% EtOH (30 s) (care was taken not to allow sections to dry out), prior to 3 washes in PBS. Sections were then blocked with Fc Receptor Blocker (Innovex #NB309) for 15 min at RT and then with Background Buster (Innovex #NB306-50) for 15 min at RT, prior to washing once with PBS. Sections were then coated with 200 μL of primary antibodies (1:500 diluted 1:500 in 10% Background Buster: PBS) and incubated for 1 h at 37 °C or overnight at 4 °C, prior to 2 washes of 10 min each with PBS. Sections were then stained with 200 μL of secondary antibodies (1:1000 diluted in 10% Background Buster: PBS) and DAPI (10 μg/mL) for 30 min at 37 °C. Finally, the slides were washed 2 times with PBS, 15 min each, and refixed with 4% PFA for 10 min. Sections were mounted using Immu-Mount (Epredia) and #1 coverslips (Electron Microscopy Services), sealed with nail polish, and stored at −20 °C until they could be imaged.

### IF measurement of DSBs in tissues
Methods are the same as for IF detection of proteins except an antibody for γH2AX[49] or 53BP1[50,51] was used (Table S1). Tissue sections were co-stained with DAPI, an antibody to NeuN, and antibodies to γH2AX. DSBs in neurons were detected as co-staining of γH2AX and NeuN. Quantification of IF staining intensity for γH2AX was determined from 50 randomly selected cells of each type within the tissue sections or in cells.

### Quantification of DSBs by CometChip assay in cells from tissue
Samples of young and old mouse brain regions (CBL, CTX, STR, HIP) were prepared before use for the Comet assay. A young (15 weeks) and an old (100 weeks) C57BL/6 mouse were sacrificed and 2–3 mm samples of the four brain regions of interest were collected, flash-frozen, and stored in liquid nitrogen until use. Samples were minced with scissors in 30 μL of buffer (HBSS, 20 mM EDTA, 10% DMSO) on ice. Next, 400 μL of buffer was added (200 μL for the HIP region, as it usually has the least cells). The mix was added on a 40 μm filter-top tube and centrifuged for 3 min at 150 × g and 4 °C. The cell concentration for each region was estimated by mixing 10 μL of each sample with 1 μL of 100x SYBR Gold (Invitrogen, Cat no. S11494), pipetting 10 μL on a hemocytometer (Hausser Scientific, PA, USA) secured on a glass slide and counting cells at 5× magnification using an inverted LED fluorescence motorized microscope (Zeiss LSM 710 microscope, Carl Zeiss Microscopy, GmbH, Germany). Every sample was then brought to 300,000 cells/mL in 25 μL with PBS and used for the Comet assay, as described for glial cultures.

### Brain sample preparation for the tissue Comet assays
Samples of young and old mouse brain regions (CBL, CTX, STR, HIP) were prepared before use for the Comet assay. Young (15 weeks) and old (roughly 100 weeks) C57BL/6 mice were sacrificed and

2–3 mm2 samples of the four brain regions of interest were collected, flash-frozen, and stored in liquid nitrogen until use. Samples were minced with scissors in 30 μL of buffer (HBSS, 20 mM EDTA, 10% DMSO) on ice. Next, 400 μL of buffer was added (200 μL for the HIP region, as it usually has the least cells). The mix was added on a 40 μm filter-top tube and centrifuged for 3 min at 150 × g and 4 °C. The cell concentration for each region was estimated by mixing 10 μL of each sample with 1 μL of 100x SYBR Gold (Invitrogen, Cat no. S11494), pipetting 10 μL on a hemocytometer (Hausser Scientific, PA, USA) secured on a glass slide and counting cells at 5× magnification using an inverted LED fluorescence motorized microscope (Zeiss LSM 710 microscope, Carl Zeiss Microscopy, GmbH, Germany). Every sample was brought to 300,000 cells/mL in 25 μL with PBS and used for the Comet assay.

### Quantification of DNA damage using the Comet assay
Alkaline and neutral Comet assays were performed to quantify the amount of DNA damage and DSB in different young or old mouse brain regions (CBL, CTX, STR, HIP). In brief, 25 μL of each sample at 300,000 cells/mL was mixed with 250 μL molten low-melting agarose (R&D Systems, Cat no. 4250-050-02). 40 μL of this mixture was pipetted onto a 3-well FLARE slide (R&D Systems, Cat no. 3950-075-02) and spread with the pipette tip. Slides were placed in the dark for 10 min at 4 °C, and then immersed in Lysis Solution (R&D Systems, Cat no. 4250-010-01) overnight at 4 °C. For the Neutral Comet, slides were immersed in cool neutral buffer (0.1 M Tris, 0.5 M Sodium Acetate, pH 9) for 30 min, before electrophoresis was performed at 21 V for 45 min at 4 °C in 850 mL of neutral buffer. Slides were put in DNA Precipitation Solution (1 M Ammonium Acetate in 95% EtOH), then 70% EtOH for 30 min each. For the Alkaline Comet, slides were immersed in the alkaline buffer (200 mM NaOH, 1 mM EDTA, pH > 13) for 20 min in the dark, before electrophoresis was performed at 21 V for 30 min at 4 °C in 850 mL of alkaline buffer. Slides were put twice in distilled water and once in 70% EtOH for 5 min each. For both Comet assays, slides were dried at 37 °C for 10–15 min. 100 μL 1x SYBR Gold (Invitrogen, Cat no. S11494) were placed onto each well for 30 min in the dark, before being rinsed with distilled water. Slides were dried completely at 37 °C, and fluorescence 5 × 5 tilescan (0.6 zoom) images of the comets were captured at 5× magnification using an inverted LED fluorescence motorized microscope (ZEN 2.1 SP3 FP3 (black) Zeiss LSM 710 microscope, Carl Zeiss Microscopy, GmbH, Germany). Comet images were analyzed using Trevigen comet analysis software (R&D Systems, MN, USA). We scored at least 200 cells per sample. The average value of DNA in the comet tail (%) and tail moment were used as the parameters for estimating basal DNA damage and DSB levels using alkaline and neutral Comet assays, respectively.

### Preparation and culturing of primary glia from brain regions
Intact brains were collected from day 1–3 newborn pups (P1-3) of male C57BL/6J mice. Brain regions (hippocampus, cerebellum, cortex, and striatum) pooled from 4–8 pups were isolated in a solution of HBSS supplemented with 1 mM L-glutamine, 1 mM sodium pyruvate and 1x Non-Essential Amino Acids. These tissue suspensions were digested in 5 mL 0.025% Trypsin-EDTA (Gibco 25300056) for 20 min at 37 °C with gentle rocking. Tissue pieces were pelleted (5 min, 300 × g, room temperature (RT)) and then gently triturated 20–30 times in pre-warmed astrocyte media (Neurobasal A base media (Thermo-Fisher #10888022), 10% FBS (JRS 43635), 2% B27 Supplement (Thermo-Fisher #1504044), 25 mM glucose, 2 mM sodium pyruvate, 2 mM GlutaMax, 1x non-essential amino acids (Quality Biologicals 116-078-721EA), 1x antibiotic/antimycotic (Gibco #15240062) using a 5 mL pipet, to dissociate into cells. Each cell suspension was tested for mycoplasma and negative cultures were passaged 3 times to expand cell number. Each passage was cultured for 6–10 days (at 37 °C, 5% CO$_2$) with media

exchanges every 2–3 days. At the end of three passages (roughly 25 days in culture), some cultures developed traces of mycoplasma but the transfection results of these cultures did not differ significantly from the negative cultures. Astrocyte cell purity and homogeneity was established by immunofluorescent analysis using anti-GFAP antibody-Cy3 conjugate (Abcam ab49874).

## Quantification of CometChip assays for glial cells

Neutral CometChip assays were performed to quantify the amount of DSB in the primary glia isolated from different mouse brain regions (CBL, CTX, STR, HIP). Assays were performed in three different biological replicates (on three different days from the same batch of cells used for FM-HCR assays). The CometChip preparation and protocol were carried out as described[52,85–87]. All chemical reagents used for CometChip assays were obtained from Sigma or VWR unless otherwise stated. In brief, 10,000 cells per 100 µL/well were loaded for each cell type in 96-well plates for 60 min at 37 °C and 5% $CO_2$. After settling by gravity, excess cells were aspirated, and the chips were rinsed with 1X DPBS (Gibco, Cat no. 14-040-133). Then, 1% (w/v) molten low-melting agarose (Gold Biotechnology, Cat no. A-204-25) in 1X DPBS was overlaid onto the chip and allowed to gelate for 2 min at room temperature followed by 2 min at 4 °C. The neutral chip was submerged overnight at 43 °C in pre-warmed neutral lysis solution (2.5 M NaCl, 100 mM $Na_2EDTA$, 10 mM Tris, 1% N-lauroyl sarcosine, pH 9.5 (TCI America, Cat no. S05975G) with 0.5% Triton X-100 and 10% DMSO). After overnight incubation, for the neutral comet chip, DNA unwinding was performed with neutral buffer (2 mM $Na_2EDTA$, 90 mM Tri's base, and 90 mM boric acid in distilled water) at 4 °C for 60 min. Electrophoresis was carried out in same DNA unwinding buffer at 4 °C for 60 min at constant voltage of 0.6 V/cm and 6 mA for the neutral chip. Following the electrophoresis, the chips were neutralized using neutralization buffer [0.4 M Tris-HCl buffer, pH 7.5] and stained using 1X (0.01% (v/v) of 10,000X stock) SYBR gold nucleic acid gel stain (Invitrogen, Cat no. S11494) for 15 min at room temperature. Fluorescence images of the comets were captured at 10× magnification using an inverted LED fluorescence motorized microscope (Zen 3.2 pro blue edition, Zeiss Apotome 2, Carl Zeiss microscopy, GmbH). Comet images were analyzed using Trevigen comet analysis software (R&D Systems, MN, USA). At least 50 cells were scored per biological replicate. The median tail length (µm) was used as the parameter for estimating basal DSB levels in the neutral CometChip assay[52,53,85–87].

## Transfections in primary glia cultures

Primary mouse glia was used live or thawed and cultured in pre-warmed DMEM high glucose medium supplemented with 20% fetal bovine serum, GlutaMax, non-essential amino acids, plasmocin, normocure, and penicillin/streptomycin. For transfection experiments, 0.05 million cells were seeded per well in 12-well plates and allowed to attach overnight. All cells were passages once to achieve about 80% confluency. On the day of transfection, cell culture medium was changed to 1 ml OPTI-MEM, and the cells were transfected with 1 µg plasmid cocktails using lipofectamine 3000 (Thermofisher Scientific) according to the manufacturer's protocol. The transfection medium was changed to cell culture medium at 4 h post transfection. At 24 h post transfection, the cells were washed with PBS, trypsinized, and filtered through a 40 µm strainer cap tubes for flow cytometry analyses.

## FM-HCR reporter vectors

All reporter plasmids used for FM-HCR were pMax vector-based. They were engineered with site-specific DNA lesions as previously described[56–58]. For NHEJ reporter, pMax BFP with a ScaI recognition site (BFP-ScaI) was linearized with ScaI restriction and purified using phenol-chloroform extraction and ethanol precipitation method. For the NER reporter, pMax mPlum was irradiated with UV-C light at 800 J/m² and purified using ethanol precipitation method. The

mPlum UV reporter was validated by analytical digest with T4PDG enzyme. For BER, the pMax GFP THF reporter was used to report long-patch base excision repair activity, and, for MMR, the pMax GFP A:C MMR reporter was generated using the described protocol[66,68]. Briefly, the base plasmid pMax-GFP C289T was nicked with Nt. BspQ1, digested with ExoIII to generate ssDNA, annealed with an oligo, that contains the mismatch. Subsequently, the oligonucleotide was extended by an overnight incubation with DNA polymerase and ligase, cleaned up with T5Exo digestion, precipitated with PEG8000 solution, and purified using phenol-chloroform extraction and ethanol precipitation. The GFP AC MMR reporter was validated in TK6 MSH2[-/-] and WT cells.

HR activity was reported using a new BFP HR assay, which was designed with reference to the reported pCX-NNX-GFP HR assay[85]. Briefly, three gene blocks were designed to generate a wild-type plasmid (2Nt-BFP), a donor plasmid (2Nt-D3BFP), and a plasmid (2Nt-D5BFP) with a ScaI restriction recognition site. Each gene block contains an NheI recognition site at the 5' end and a SacI recognition site at the 3' end. Adjacent to the NheI recognition site are two recognition sites for Nt. BspQ1 (2Nt) that flank the recognition sites for MluI and ScaI, and all of which are upstream of the Kozak consensus sequence. For 2Nt-D5BFP, the last four base pairs of the Kozak sequence and the first 19 base pair from the 5' end of BFP were deleted to generate a novel Pst1 recognition site and a truncated BFP sequence that renders the protein encoded by the plasmid non-fluorescent. For 2Nt-D3BFP, 99 base pairs were deleted from the 3' end to generate a truncated BFP protein that renders the protein expressed by the plasmid non-fluorescent. The 2Nt-BFP, which serves as the positive control, is engineered to have a full-length BFP sequence, and thus is fluorescent. To generate the three HR reporters, pMax vector and gene blocks were digested with NheI and SacI and purified using gel extraction (Monarch DNA gel extraction kit, NEB) for the linearized vector and column purification (Monarch PCR and DNA cleanup kit, NEB) for the gene blocks. The gene blocks were then cloned into the linearized pMax vectors and amplified in DH5α competent cells. Putative positive clones were selected by kanamycin resistance and validated by sequencing using pMax reverse primer. Individual reporters were amplified in 1 L LB medium with kanamycin and extracted using PureLink endotoxin-free giga plasmid purification kit (Thermofisher Scientific). For generating the linearized 2Nt-D5BFP for HR assay, 2Nt-D5BFP was linearized with PstI and purified using phenol-chloroform extraction and ethanol precipitation method. Fluorescence of each vector was tested in cell lines using flow cytometry analyses and the recombinogenic events between the linearized 2Nt-D5BFP and 2Nt-D3BFP were validated in HR deficient cell lines.

## FM-HCR reporter cocktails

Three reporter cocktails were prepared for FM-HCR analyses. Damaged reporter cocktail A is composed of 100 ng of PstI-linearized 2Nt-D5BFP (HR), 100 ng of pMax GFP THF (BER), and 100 ng of pMax mPlum as transfection control. Damaged reporter cocktail C is composed of 100 ng of GFP_AC reporter (MMR) and 100 ng of pMax_mPlum as transfection control. A final amount of 2Nt-D3BFP was added to each damaged plasmid cocktail so a total of 1400 ng plasmid DNA was used for transfection. A single undamaged reporter cocktail is composed of 1100 ng of 2Nt-D3BFP, 100 ng of each of BFP_ScaI, pMax_GFP, and pMax_mPlum.

## FM-HCR analysis

Primary glia were transfected with sufficient efficiency (4–8%) to afford a robust analysis of DNA repair capacity by flow cytometry analyses. All transfected cells were analyzed by flow cytometry (Attune NXT Flow cytometer, Thermofisher Scientific) at 24-h post transfection. Data analyses were performed as previously described[56–58] to determine the DNA repair capacity of the primary glia. Briefly, for each transfection

($n = 5$), the product of the fluorescent positive cell counts and mean fluorescence intensity of the positive cells in each gate was normalized to that in the gate for transfection control. Activity of each DNA repair pathway was reported as % reporter expression, which is the ratio between the normalized reporter expression in the damaged and the undamaged cocktail multiplied by 100. The activity level of each DNA repair pathway studied is positively correlated with the % reporter expression. FM-HCR assays report independent repair mechanisms and absolute reporter expression depends on factors specific to each assay precluding statistical comparisons between pathways. Thus, % reporter expression approximates the percentage of reporter plasmids that have been repaired by the pathway of interest, this approach provides a rough estimate of the relative activity of multiple pathways based on this metric. FM-HCR assays report independent repair mechanisms and absolute reporter expression and depend on factors specific to each assay, precluding statistical comparisons between pathways. The FM-HCR assays are highly validated, robust reporters of DNA repair activity of cells and reflected activity of the line. For example, using the same FM-HCR assays, a panel of human lymphoblastoid cell lines from apparently healthy individuals consistently yielded around 2% reporter expression for HR, as was observed in the glia, but had greater than 30% reporter expression for NHEJ[56–58]. At the same time, expression for the BER reporter was high (~40%) in mouse brain glial cells and in human lymphoblastoid cell lines[56–58].

## DNA oxidation assay
Three male C*57BL/6J* mice were assessed from both young (7–10 weeks old) and old (70–90 weeks old) ages, as indicated, referred to as wild-type (WT) mice (Jackson Labs cat#000664). Briefly, the CBL, STR, HIP, and CTX were collected from each animal and the DNA was isolated from 25 mg of tissue using DNeasy Blood and Tissue Isolation kit (according to the manufacturer's protocol). The DNA was quantified (using a Nanodrop instrument, Thermo-Fisher) and 25–30 µg DNA was diluted in 45 µL distilled/deionized water. The genomic DNA was sheared at low power using a sonicator for 30 s on ice. Double-stranded DNA was heat denatured at 95 °C for 5 min before being snap cooled on ice for 10 min. DNA was then digested with 150U of Nuclease P1 (New England Biolabs) in 1x P1 buffer (New England Biolabs) for 30 min at 37 °C. The reaction was stopped by heat inactivating for 10 min at 70 °C and addition of 1 µL 1 M Tris-Cl pH 8.0. The nucleotide solution was further digested with 5U Quick CIP Calf Alkaline Phosphatase (New England Biolabs) for 30 min at 37 °C, before being heat inactivated for 5 min at 80 °C and snap cooled on ice until it was used in the assay. Oxidized DNA/RNA was then quantified following manufacturer's protocol (DNA/RNA Oxidative Damage ELISA Kit, Cayman Chemicals #589320).

## XJB-5-131 treatment
XJB-5-131 (gift form P Wipf (University of Pittsburgh) was synthesized[62] and treatment protocols were as previously descriend[60–63]. Lyophilized, powdered XJB-5-131 was reconstituted in DMSO at a concentration of 1 µg/µL. These samples were aliquoted and kept at −80 °C. On the day of injection, the XJB-5-131 solution was mixed with 0.2 µm filtered and pre-warmed PBS (100 °C) to reach a final concentration of 2 mg/kg mouse body weight in 200 µL solution. This was heated for 10 s and the solution (200 µL) was injected, within 30 min of preparation, by intraperitoneal injection (IP). Administration started at 60 weeks of age and continued three times per week for the length of the study. Vehicle treatments were identical except that XJB-5-131 was replaced by filtered PBS.

## Peroxide treatment in cells
Mouse Embryonic Fibroblast cell line (NIH3T3, ATCC Cat# CRL-658) were grown overnight in 8 well microwell slides (Ibidi Cat# 80826). Experiment 1 cultures in Fig. 9 were peroxide treated (100 µM) with

and without XJB-5-131 (200 µM) for 30 min. After all treatments, the cells were washed briefly with PBS and fixed in 4% Paraformaldehyde for 15 min prior to immunofluorescent staining.

## Transfection and targeting of Cas9 and Cas9D10A
Cells were transfected with plasmids expressing the hCas9-D10A nickase or Cas9. DNA using Lipofectamine 3000 (ThermoFisher Cat#L3000001) following the manufacturers protocol. Briefly, DNA (500 ng total per well) was added along with 0.5 µL P3000 reagent and 0.5 µL Lipofectamine 3000 reagent after a short (10 min) mixing and pre-incubation. The transfected DNA was a combination of guide RNA plasmid (recognition sequence 5' CCATATTCCACGTCCTACAG 3'), hCas9D10A nickase (2) (Addgene Cat#41816) or Cas9 (pCDNA3.3-TOPO-T7-hCas9) (Addgene Cat #161876) in the presence of Doxycyclin (2 µg/mL), with or without XJB-5-131 (200 µM). After 6 h, they were washed with PBS and fixed in 4% Paraformaldehyde for 15 min prior to immunofluorescent staining.

## Statistical analysis
Statistical analysis is reported as appropriate in each Figure. For all box and whisker plots, the boxes represent 50% of all data points with the line indicating the median value. The remaining 50% of datapoints are in the whiskers, distributed as the 25% maximal values above the box and 25% minimal values below the box. For CometChip assays, graphical representations were expressed as Mean ± SD and one-way ANOVA followed by Tukey's multiple comparisons test was performed using GraphPad PRISM version 9.5.1 for Mac, GraphPad Software, San Diego, California, USA. $P < 0.05$ was considered as statistically significant using ANOVA models with post hoc multiple comparison tests by GraphPad PRISM (GraphPad Software, LLC).

## Reporting summary
Further information on research design is available in the Nature Portfolio Reporting Summary linked to this article.

## Data availability
The data supporting the findings of this study are available from the corresponding authors upon request. Source data for the figures and Supplementary Figs. are provided as a Source data file. Reagent used are presented in Supplementary Table 2. Antibody testing results are presented in Supplementary Table 3. Source data are provided with this paper.

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

## Acknowledgements

This work was supported by NIH (R01 NS060115 and R01 GM066359 to C.T.M. and U01ES029520 to Z.D.N.).

## Author contributions

All authors participated in discussion of the data, data analysis, and contributed to the manuscript writing. A.A.P. Prepared cells and tissue samples for all experiments, performed and analyzed the single-cell experiments throughout for protein expression (Fig. 3, Supplementary Figs. 2, 3, 4, 5, 6, 7 and 8, and DSB analysis in tissues (Fig. c, d) and in cells (Fig. 5c), IF of mouse tissues throughout the manuscript (Figs. 2, 3, 4 and 9); 8-oxoG assays (Fig. 7b, c); XJB-5-131 in vivo analysis (Fig. 8c), and Cas10DA and Cas9 targeting and XJB-5-131 analysis in cells (Fig. 9). A.C. Performed and analyzed the FM-HCR assay (Fig. 6). J.-H.Y. Performed and analyzed the western analysis of protein expression (Fig. 1 and Supplementary Fig. 1); and constructed the vector for MajSat guide RNA (Fig. 9). S.M.T. Performed and analyzed the Neural and Alkaline CometChip in glial cultures (Fig. 5a, b). L.B. Performed and analyzed the neutral and alkaline CometChip analysis in mouse tissue (Fig. 4d, e, Supplementary Fig. 5, Fig. 7e, f). Z.D.N. Overall direction of FM-HCR and CometChip in glia cultures. Developed FM-HCR analysis and tools and participated in critical evaluation of the manuscript. C.T.M. Overall experimental design and critical evaluation of the project; wrote the manuscript with contribution from all authors.

## Competing interests

The authors declare no competing interests.
