## [Peer Review File · Nature Communications]

Base excision repair and double strand break repair cooperate to modulate the formation of unrepaired double strand breaks in mouse brainREVIEWER COMMENTS

Reviewer #1 (Remarks to the Author):

Polyzos et al. report an analysis of brain region-specific and cell type-specific (Neuron vs Glia) levels of DNA repair factors in mice. They generally found increased DNA repair factors from major DNA repair pathways in Neurons compared to Glia. They also found increased DNA lesions as measured by γ -H2AX in neurons compared to glia. Finally, they show some evidence for SSB to DSB conversion. Overall, while the authors tried to address an important and emerging question of DNA repair in the brain cells, there are several concerns with this study. The data is confusing, with few replicates performed, and some datasets where conclusions were made even when there is no statistical significance. Also, some figure panels are replicated. Finally, the data is inadequate to support the model proposed, especially the part where DSBs act as damage sensors. Given these concerns that are elaborated below, I don't recommend publication of this manuscript at Nat Comm.

Major

1) The Western Blot (WB) and Immunofluorescence (IF) have opposing results, where CBL has high DNA repair factors according to WB and low according to IF. The authors suggest that the reason for this is that neurons have higher levels of DNA repair factors and CBL is enriched for neurons. However, the CBL neurons seem to have almost the same level of DNA repair factors as the glial cells (Figure 3b,c, Figure S4). Therefore, this argument doesn't make sense and an alternate explanation or additional data should be provided.

Also, I don't understand the conclusion being drawn in 206-210, where the author states that the neurons, independent of the surrounding context (CBL or STR), have high DNA repair factors when the data clearly shows that CBL neurons are drastically different from other regions in terms of their DNA repair factor levels.

2) WB significance in Figure 1 is calculated based on $n=2$ which is not adequate and should have more replicates.

3) Figure 3b, c (young mice) is an exact copy of Figure S3 a, b (old mice). Also, no statistics are shown.

4) Figure 3a shows representative IF images, but it is unclear from which regions. Please label this in the figure. Additionally, it will be useful to provide representative IF images for all regions in a supplementary figure.

5) In Figure 4b, only γ -H2AX is quantified. Please provide quantification for 53BP1 as well.

6) The results of the FM-HCR assay not reaching any statistical significance are concerning for drawing conclusions. The authors say that's due to glial cultures being of different passages. Using culture systems should allow for maximum control over variables and reproducibility relative to animals.

7) Fig 7 e-h is exactly repeated from Figure 1. Traditionally one can just refer back to the figure instead of repeating it as a new figure. Also in Figure 7g, CBL to STR comparison is marked as significant when it is not according to Figure 1, and also based on the distribution of the points. Related to this, what is the difference between Figure 4b γ -H2AX quantification

(age of the mice not labeled) and Figure 7i γ -H2AX quantification of mice at 7wks.

8) Please provide representative images of γ -H2AX staining for figures 7i,j and 8c.

9) In lines 233-240 and figure S5, the authors show that DSB repair protein levels correlated to DSB lesion levels. But in Fig 7 the authors are invoking the opposite suggestion (lines 322-325) that a decrease in Ku70 increases DSBs in aging. If the latter suggestion is true then glial cells that have low DSB repair proteins (Fig 3c) should have the highest amount of DSB lesions.

10) The only piece of evidence that shows possible SSB conversion to DSB here is the reduction of DSBs after treatment with the antioxidant drug. This one experiment is not sufficient to make a major claim in the abstract that “We propose a dynamic equilibrium model in which non-dividing brain cells use DSBs as direct damage sensors by reversibly regulating the response of brain cells to oxidative stress.”

Experiments required to conclusively show the conversion dynamics include:-

- Showing SSB to DSB conversion, with alkaline and neutral comet assay to directly test the lesions (instead of just γ -H2AX and 8-oxo-G)
- An increase in oxidative stress increases SSB to DSB conversion
- Showing these in neurons as well, since the claim is for all non-dividing brain cells

11) Finally, the idea that cells use DSB, which is a graver threat to genome integrity than oxidative DNA damage, as a sensor for oxidative damage requires a high level of proof.

Minor:

1) The age of mice in Figure 3 is not labelled (I believe 7 wks). Please label age in all figure panels. Also, in line 191, the text says 70 wks and the figure says 75 wks.

2) Lines 189-191: “Neurons (N) expressed DNA repair proteins of all five DNA repair pathways at higher levels than in young (Fig.S2a,b and Fig. 3a-c) and old (70wks) (Fig. S3a,b) mouse brains.”

Should it be “Neurons (N) expressed DNA repair proteins of all five DNA repair pathways at higher levels than in Glia (G) in young (Fig.S2a,b and Fig. 3a-c) and old (70wks) (Fig. S3a,b) mouse brains.”

Reviewer #2 (Remarks to the Author):

This manuscript presents a correlation study of 4 DNA repair pathways in brain, via measuring the levels of some proteins in the pathways, and employing transfected reporter probes in cultured primary cells as measures of activity. The results are interpreted to suggest that neurons and glia somehow respond to the SSB/DSB ratio in a kind of homeostasis. There are some significant technical issues, and limitations of what can be concluded, as detailed below. The writing needs some work, too.

SPECIFIC POINTS

Introduction: Pretty long, and a bit repetitive. It would be improved by being condensed significantly. In fact, other sections of the manuscript could be productively edited down as

well.

Fig. 1 & legend lines 704-705: APE1 may not be the best BER indicator, as its levels can vary a lot, in ways that are poorly understood. That the protein is also implicated in multiple other functions further complicates it as a suitable BER marker. At the same time, the differences observed are rather small.

Fig. 2: How sharply defined are the two populations? Antibody staining can be quite variable.

Fig. 4d: The tail moments are extremely small; I am not sure that this is really within the precision of the method. Justify this.

P. 12/lines 280-281: It is very important to state the range of observed transfection frequencies observed in the experiments.

Fig. 5d/Line 290-291: What constitutes a “trend”? It looks to me like the relative activities of different pathways as measured by this assay are very similar in the 4 sampled populations. But that’s as likely to be a by-product of isolation and culturing, so it doesn’t merit saying much more, in my opinion.

Fig. 6 & associated text: SSB are a part of normal cell function, which ought to be noted here. Are the SSB numbers corrected for DSB? That seems appropriate so that you are not counting things twice. If you didn’t do that, does applying such a correction, say $\text{NetSSB} = \text{ObsSSB} - 2(\text{ObsDSB})$, change the conclusions?

Lines 329-330: As noted, oxidative damage includes many things, and the same processes generate other DNA damage that could lead to DSB. Free radicals hit many other targets, too, so the effect need not even depend on BER. Transcription in particular is a significant source of SSB (Alt & Schwer DNAR 71:158; 2018).

Discussion: Does the proposed BER/DSBR imbalance predict that glia and neurons are less sensitive than other cell types to the DSB-triggered checkpoints? That might have implications as much for cancer etiology as for treatment.

Lines 367-368: “If and how a DSB formation-suppression mechanism might occur” is kind of central to the thrust of the paper. But it’s hard to discern a regulatory mechanism in the text, and it is not known whether the differences are causes or effects.

Line 402: Among the practical considerations is that the primary cell cultures were (apparently) not synchronized. So the transfection efficiency, the location of the probe molecule within the cell, and the activity of different DNA repair pathways are all likely to vary in an uncontrolled fashion and substantially increase the resulting noise.

Lines 409-411: Note that the generation of some endogenously-generated DNA lesions is likely to vary quite a bit among cell types and tissues, with further variations arising from the physiology-dependent pathways (e.g., free radicals).

Fig. 9: I don’t find this very useful; it doesn’t illuminate anything not already clear in the text.

MINOR POINTS

General: there are quite a few small errors/typos, indicating that better proofreading was needed. The comments below include a couple of examples, but there are more.

Abstract, lines 32-35: This sentence is opaque, probably as a result of trying to cram too many things into it.

Line 58: "non-diving"

Lines 65-66: This repeats a common misinterpretation. 8-oxo-dG is formed at rates similar to many other oxidative lesions, and it constitutes just a few percent of the total. It is simply better known and somewhat better understood.

Results, lines 220-222: This is not a sentence.

Lines 321-322: The sentence could be misinterpreted to suggest that you to mean that certain genome locations always have an SSB. If you mean that, say so, otherwise revise.

Lines 343-345: "Although SSBs can arise from BER, NER/TCR or MMR, XJB-5-131 alone could block the level of DSBs in neurons at older age." It's not clear what the meaning of this sentence is.

Discussion, final paragraph: Many other tissues are also relatively quiescent, so would the same process as proposed for neurons also occur in those? What evidence might there be for that?

Reviewer #3 (Remarks to the Author):

This manuscript aims to show that DNA damage repair mechanisms vary in different regions of healthy mouse brains, with BER being the most common method. Reactive oxygen species accumulation causes DSBs. However, the results lack evidence and the sample size is too small ($n=2$) to be convincing.

1. The study's authors demonstrate that DNA repair mechanisms are region-specific in the brain by detecting the expression of specific critical proteins related to DNA damage repair pathways through immunoblotting (Fig. 1 and Fig. S1). However, the figures only provide data from two mice ($n=2$) and do not indicate the intrinsic reference protein used in the assay. Therefore, it is necessary to present multiple samples ($n>3$) to validate the results.

2. The authors also conclude that the expression of proteins related to the DNA damage repair pathway is significantly higher in neurons than in glial cells in the same mouse brain region. However, the immunofluorescence figures shown in the manuscript lack co-staining results of DNA damage repair proteins and NeuN, which is crucial in determining the cell type of the displayed cells.

3. The authors conclude that the primary method of DNA damage repair in the cerebellum and hippocampus is non-homologous end joining (NHEJ), based on the immunofluorescence staining results of 53BP1 in neurons. To support this conclusion, the authors must provide immunofluorescence figures of other DNA damage repair pathway proteins.

4. In Fig. 4a, the authors show that γ H2AX staining is region and cell-specific in the brain. However, the immunofluorescence picture only shows one brain region, and this set of figures does not provide co-staining results with markers of neurons or glia.

5. The authors address that the accumulation of reactive oxygen species is the main reason for the formation of DNA double-strand breaks (DSBs) in the brain (Fig. 8). Nevertheless, the authors believe that DSBs in normal brain tissue are transformed from single-strand breaks (SSBs). This conclusion cannot be approved based solely on the antioxidant drug XJB-5-131 injection assay.

6. Page 10, 2nd para, "However, the opposite was true; the level of DSB lesions (measured by γ -H2AX staining) (Fig. 4b) was directly proportional to the expression level of Ku70 and Ku80 in the same cell type (Fig. S5). No data for Ku70 is shown.

Point-by-point.
Rev. 1
Major

(1)(a) The Western Blot (WB) and Immunofluorescence (IF) have opposing results in the CBL, where it has high DNA repair factors according to WB and low according to IF. ..(b) I don't understand the conclusion being drawn in 206-210 ... neurons, independent of the surrounding context (CBL or STR), have high DNA repair factors when the data clearly shows that CBL neurons are drastically different from other regions in terms of their DNA repair factor levels.

Western analysis was used to establish that brain cells of four regions express DNA repair proteins needed to carry out repair. Indeed, all brain region expressed DNA repair machinery at all ages, but the region themselves had different features, which made accurate comparisons difficult. CBL was an outlier. It differed in cell density, and cell type composition (Fig. 2) and significantly in cell size, which influences nuclear protein concentration⁴⁸ (Lanz *et al.*, *Mol Cell*, 2022). Although normalized to total protein for loading, the cerebellar cells are very small and had five times more cells per unit total protein than other regions (see new Fig. 1c, 350ng/ml). When the samples were adjusted for cell number, the expression level among regions became more similar. The CBL also differed in composition from other brain regions. Neurons and glia differ in size; CBL was 80% neurons while the other three regions were around 20% neurons (Fig, 2). Therefore, we used the western analysis to show that brain cells of four regions express DNA repair proteins needed to carry out repair—but quantitative comparisons were made at the single cell level (in Fig. 3), which were used in the rest of the paper. We have revised the text to make this clear.

(c) WB significance in Figure 1 is calculated based on n=2 which is not adequate and should have more replicates.

There were no instances where the samples were measured less than 3 times. All the WB data was derived from samples comprising 3-6 pooled animals which reflect the average of 3-6 biological replicates in each sample. We made two such pooled samples (hence n=2), and they were analyzed in triplicate. The SDS-PAGE gel for one of the triplicates is included in Fig. 1. The methods and legends have been checked for accuracy.

(d) Figure 3b, c (young mice) is an exact copy of Figure S3 a, b (old mice). Also, no statistics are shown.

Thank you for pointing this out. The correct data for old animals are included as new Fig. S3 and statistics are included in both Fig 3 and Fig. S3. The 7 wk and 75 wk expression profiles are very similar but not the same. In the revised text, we explicitly state that expression profiles do not change significantly with age.

(2) *Figure 3a shows representative IF images, but it is unclear from which regions. Please label this in the figure. Additionally, it will be useful to provide representative IF images for all regions in a supplementary figure.*

The representative images of protein expression were taken from 7 wk CTX, which is now labeled in the revised manuscript. In the revision, we have added 3 more images of five pathway proteins in the cortex to establish the consistency of cell type distribution for glia and neurons in all five pathways (Fig, S2), Additionally, we have included new images of protein expression for a second protein (MSH6) in all four brain regions in new Fig. 4.

(3) *In Figure 4b, only γ -H2AX is quantified. Please provide quantification for 53BP1 as well.*

In new Fig. 4b, we include images for γ H2A-X staining and 53BP1 in the same cell (in separate channel images) in addition to the overlay. Additionally, we provide quantified γ H2A-X and 53BP1 images in the same cells in side-by-side images in three cell experiments (new Fig. 7c, e, i). We have also backed up these findings using a third experiment—comet assays—as a direct measure to support the DSB measurements from the γ H2A-X and 53BP1 markers (new Fig. 4d,e and new Fig. 6e,f).

(4) *The results of the FM-HCR assay not reaching any statistical significance are concerning for drawing conclusions... Using culture systems should allow for maximum control over variables and reproducibility relative to animals.*

We did not see consistent passage-dependent effects on repair activity, but some variability was introduced in the biological replicates, all of which were isolated from a different set of animals. However, old panel 5c showed that any given DNA repair pathway had about the same activity in all four brain regions. The old panel 5c was eliminated. We have revised the paper with a new Fig. 5c where we plot the same data side by side as relative repair pathway activity per region. As is obvious by the log₁₀ scale axis, the pathway activity is significantly different in each brain region: BER activity is high and DSBR activity is low in all regions. The methods are well-established.

- Nagel, Z. D. *et al.* Multiplexed DNA repair assays for multiple lesions and multiple doses via transcription inhibition and transcriptional mutagenesis. *Proc Natl Acad Sci U S A* **111**, E1823-1832 (2014).
- Pieltz CG, Pecen TJ, Lavery DJ, Nagel ZD. Large-scale preparation of fluorescence multiplex host cell reactivation (FM-HCR) reporters. *Nat Protoc.* 2021 Sep;16(9):4265-4298.
- Nagel ZD, Beharry AA, Mazzucato P, Kitange GJ, Sarkaria JN, Kool ET, Samson LD. Fluorescent reporter assays provide direct, accurate, quantitative measurements of MGMT status in human cells. *PLoS One.* 2019 Feb 27;14(2):e0208341.

(6) *(a) Fig 7 e-h is exactly repeated from Figure 1. Traditionally one can just refer back to the figure instead of repeating it as a new figure.*

We have removed Fig. 7e-h from revised Figure 7.

(b) What is the difference between Figure 4b γ -H2AX quantification (age of the mice not labeled) and Figure 7i γ -H2AX quantification of mice at 7wks.

Quantification of γ H2AX in Figure 4b and in Figure 7i are the same. We have removed the duplicate figure.

(7) *Please provide representative images of γ -H2AX staining in figures 7i,j*

The Figure numbering in the revised manuscript has shifted. In the revised manuscript, we

have included representative images of γ H2AX staining in new Fig. 4a.

- (8) *In lines 233-240 and figure S5, the authors show that DSB repair protein levels correlated to DSB lesion levels. But in Fig 7 the authors are invoking the opposite suggestion (lines 322-325) that a decrease in Ku70 increases DSBs in aging. If the latter suggestion is true, then glial cells that have low DSB repair proteins (Fig 3c) should have the highest amount of DSB lesions.*

Fig 7 (new Fig. 6) and S5 (new Fig. S6) are consistent. The confusion lies in the fact that there are two opposing components that contribute to the observed DSBs, whose level is the sum of those formed and those repaired. Formation (increase in DSBs) was facilitated by BER removal of oxidative damage while repair (removal of DSBs) was facilitated by DSBR machinery.

Therefore, despite a direct relationship between DSBs and the expressed DSBR machinery (Fig. S6), the expressed DSBR proteins were not sufficient to completely remove all the DSBs in any region (new Fig. 4c). New DSBs occurred in parallel to those that were lost by DSBR (Fig. S6). Thus, DSB formation (by BER excision of oxidized bases) increased with age, without accompanying changes in Ku80 expression. This is supported *in vivo* in new (Fig. 6g-i). In that experiment, we suppressed the base oxidation that led to DSB formation using XJB-5-131. Although the expressed DSBR machinery was not altered, DSBs decreased with age since DSBs could no longer be formed from BER excision of oxidized bases. We have re-written the text and added new Figures S6 to make these points more clearly.

- (10) *The only piece of evidence that shows possible SSB conversion to DSB there is the reduction of DSBs after treatment with the antioxidant drug. This one experiment is not sufficient to make a major claim that there is a dynamic by reversibly regulating the response of brain cells to oxidative stress. Experiments required to conclusively show the conversion dynamics include showing SSB to DSB conversion, with alkaline and neutral comet assay to directly test the lesions (instead of just γ -H2AX and 8-oxo-G).*

Good point. We have substantially revised the manuscript by inclusion of three new key experiment (new Fig. 7) in addition to XJB-5-131 *in vivo* analysis (Fig. 6g-i). We emphasize that XJB-5-131 is a powerful proof of principle for SSB and DSB conversion *in vivo*. Oxidation damage is repaired primarily by BER through a transient SSB intermediate. XJB-5-131 suppresses ROS and base oxidation. Since XJB-5-131 suppresses DSBs, they must also depend on oxidation *in vivo*. It is difficult for us to interpret the results in any other way than the DSBs arise from two closely spaced SSBs.

However, we performed three additional experiments and include the results in the revised manuscript together with a new Fig. 7.

- (1) In the first analysis (new Fig. 7, top panel), we prepared confluent cultures of immortalized NIH3T3 cells and treated them with peroxide to elevate ROS in the presence or absence of the ROS inhibitor XJB-5-131. If SSB to DSB conversions depended on the oxidation state, we expected that peroxide treatment would elevate DSBs, and that the rise would be suppressed in the presence of XJB-5-131. Indeed, although peroxide induces modification of single bases, the transfected cells stained robustly with γ H2A-X and 53BP1 foci evident within 30 minutes of treatment. DSB foci were suppressed in the presence of XJB-5-131 (Fig. 7b, c; +XJB-5-131, black, -XJB-5-131, gray). Thus, DSB formation

depended on base oxidation *in vitro* and appeared to arise from conversion of two SSBs that formed during removal of the oxidized bases.

- (2) In a second experiment (new Fig. 7, middle panel), we tested whether the reduction in DSBs by XJB-5-131 could occur in the absence of base oxidation (Fig, 7d). The premise being that SSB to DSB conversions would be overestimated if non-oxidative lesions contributed to the population of DSBs. To test this idea, cell cultures were prepared as in experiment 1 but, in this case, genomic SSBs were generated enzymatically by transient transfection of recombinant CAS9D10A. This mutant converts the Cas9 nuclease into a “nickase” capable of introducing only one SSB into DNA per reaction (*Mali, P., Yang, L., Esvelt, K. M., Aach, J., Guell, M., DiCarlo, J. E, Church, G. M. (2013). RNA-guided human genome engineering via Cas9. Science, 339(6121), 823-826.*). We targeted SSB formation to the major satellite DNA using “sense” Cas9D10A guide RNA that was complimentary to the site. In the transfected cells, we tested whether (1) SSBs could be converted to DSBs in the absence of oxidation and (2) whether DSB formation under these conditions could be suppressed by XJB-5-131. Although CAS9D10A induced only SSBs, the transfected cells stained robustly with γ H2A-X and 53BP1 markers within 6 hours post transfection. The results confirmed that SSB to DSB conversion had occurred. Furthermore, the IF signals of γ H2A-X and 53BP1 co-localized with the CAS9D10A RNA guide bound at the Major satellite loci, indicating that SSBs and DSBs formed at the same Major satellite target sites. Notably, the formation of CAS9D10-directed DSBs were not suppressed by treatment of cells with XJB-5-131. Collectively, the results from experiments 1 and 2 demonstrated that DSB formation depended on base oxidation, which was suppressed by XJB-5-131; SSB to DSB conversions occurred during excision of oxidative base damage.
- (3) In a third experiment (new Fig. 7, lower panel), we established whether XJB-5-131 had any impact on DSB formation itself. If XJB-5-131 inhibited DSBs directly, for example, then their reduction might be unrelated to oxidation. We repeated experiment 2 but replaced CasD10A with active Cas9 and targeted it to the Major satellite DNA using Cas9 guide RNA. The Cas9 nuclease cuts duplex DNA without a SSB intermediate or DNA oxidation. Thus, we expected that DSBs would occur under these conditions but independently of XJB-5-131 treatment. Indeed, DSBs formed in these cells, as detected by IF intensity of γ H2A-X and 53BP1 staining. Notably, XJB-5-131 treatment did not have an impact on formation of DSBs. Collectively, the results of the three complementary experiments provided evidence that excision of oxidized bases promoted SSB to DSB conversion, and DSBs could be modulated by regulating base oxidation *in vitro* and *in vivo*.
- (12) *Finally, the idea that cells use DSB, which is a graver threat to genome integrity than oxidative DNA damage, as a sensor for oxidative damage requires a high level of proof.*

Although DSBs are toxic during cell proliferation or elevated in disease states, it is now well-established that DSBs have roles in normal non-dividing brain cells and serve beneficial regulatory roles.

As summarized by Alt and & Schwer (DNAR 71:158; 2018):

“Early work for about two decades implicate DNA double-strand break (DSB) formation and repair occur as part of (normal) neuronal development. Findings emerging from recent studies of DSBs in mature, non-dividing neurons suggest important roles of DSBs in brain physiology” Also see: Álvarez-Lindo, N., Suárez, T. & de la Rosa, E. J. Exploring the Origin

and Physiological Significance of DNA Double Strand Breaks in the Developing Neuroretina. *International journal of molecular sciences* **23**, 6449 (2022).

Some specific examples.

- DSBs reversibly form in the brain of animals during exercise. *Madabhushi, R. et al. Activity-Induced DNA Breaks Govern the Expression of Neuronal Early-Response Genes. Cell* **161**, 1592-1605 (2015).
- Weber Boutros, S., Unni, V. K. & Raber, J. An Adaptive Role for DNA Double-Strand Breaks in Hippocampus-Dependent Learning and Memory. *Int J Mol Sci* **23**, 8352 (2022)
- Patel, H. P. et al. DNA supercoiling restricts the transcriptional bursting of neighboring eukaryotic genes. *Molecular cell* **83**, 1573-1587.e1578 (2023).
- Suberbielle, E. et al. Physiologic brain activity causes DNA double-strand breaks in neurons, with exacerbation by amyloid- β . *Nature neuroscience* **16**, 613-621 (2013).
- Sutormin, D. A. et al. Diversity and Functions of Type II Topoisomerases. *Acta Naturae* **13**, 59-75 (2021).
- Perhaps the most significant are new findings to suggest that DSBs can regulate themselves. DSBs induce the transcription of DSB-associated factors including NPAS4-NuA4, which initiates a novel DNA repair pathway that can prevent breaks in activated neurons. (*Pollina, E. A. et al. A NPAS4-NuA4 complex couples synaptic activity to DNA repair. Nature* **614**, 732-741 (2023).

- (13) As discussed by Alt and colleagues, “neuronal activity-induced DSBs are marked by the DSB response factors γ H2AX and 53BP1 suggesting that these promoters are recognized by the DSB repair machinery. However, how precisely these breaks are resolved in neurons is currently unclear”.

We are not sure how to interpret this comment. All fields are works in progress. Alt and colleagues (2018) reviewed some ideas about DSB repair mechanisms, based current data. However, learning how neurons correct breaks is complex and will continue to expand as new data emerge and new models are tested. Thus, the reviewer has identified the exact reason for the work. Repair protein expression, DNA repair activity and lesion level have rarely been measured together. However, by integrating these parameters, our dynamic equilibrium model provides a testable mechanism for how DSBs are sensed and repaired in normal cells: oxidative damage reversibly regulates the level of DSBs. DSBs cannot accumulate forever and maintain a normal state. In other systems, DSBs induce transcription responses in neurons as a regulatory method (See *Pollina, E. A. et al. A NPAS4-NuA4 complex couples synaptic activity to DNA repair. Nature* **614**, 732-741 (2023) “DSB-associated factors NPAS4-NuA4 regulate mechanisms that prevent breaks”). The detailed mechanism is still unknown. We propose a similar mechanism based on coupling of normal metabolic flux to DNA repair. When DSBR machinery reach a threshold, they stimulate transcriptional responses to reduce oxidative damage and the number of DSB conversions. This and other proposed models will not be solved in a single publication and will require much more analysis to fully establish. The details, however, will emerge as work continues.

Minor comments

- (1) *The age of mice in Figure 3 is not labelled (I believe 7 wks).*

Corrected in the revised manuscript (yes, 7wks)

(2) *Lines 189-191: “Neurons (N) expressed DNA repair proteins of all five DNA repair pathways at higher levels than in young (Fig.S2a,b and Fig. 3a-c) and old (70wks) (Fig. S3a,b) mouse brains.”*

Corrected in the revised manuscript. The sentence is now “Neurons (N) expressed the representative DNA repair proteins of all five DNA repair pathways at higher levels than glia (G) in both young (Fig.S2a,b and Fig. 3a-c) and old (75wks) (Fig. S3a,b) mouse brains.”

REVIEWER #2.

(1) *Introduction: Pretty long, and a bit repetitive. It would be improved by being condensed significantly. In fact, other sections of the manuscript could be productively edited down as well.*

Good comment. We have re-written and condensed the intro and discussion and productively edited other sections of the manuscript. The paper has been improved by the editing.

(2) *Fig. 1 & legend lines 704-705: APE1 may not be the best BER indicator, as its levels can vary a lot, in ways that are poorly understood. That the protein is also implicated in multiple other functions further complicates it as a suitable BER marker. At the same time, the differences observed are rather small.*

We agree. Crosstalk is becoming a general feature of DNA repair with many pathways sharing components. Thus, in Fig. 1, our goal was only to determine whether key components of each repair pathway were expressed in regions of the brain at the age tested. Quantitative comparisons with protein expression were made at the single cell level (Fig.3); the BER activity was measured independently using the FM_HRC assays in cells, where activity is measured within the context of the entire BER complex.

(3) *Fig. 2: How sharply defined are the two populations? Antibody staining can be quite variable.*

Neurons are sharply defined by whether they do or do not stain with NeuN. The monoclonal anti-NeuN antibody is very specific and bright (Guselnikova, 2015, Act Nat). Thus, cells that do (neurons) or do not (glia) stain with NeuN are well resolved. (see Fig. 2).

Guselnikova, V., & Korzhevskiy, D. E. (2015). NeuN As a Neuronal Nuclear Antigen and Neuron Differentiation Marker. Acta naturae, 7(2), 42-47.

(4) *Fig. 4d: The tail moments are extremely small; I am not sure that this is really within the precision of the method. Justify this.*

Our comet tails are quantitative. Comet images were analyzed using commercial Trevigen comet analysis software (R&D systems, MN, USA). We have revised the manuscript not only to show more extensive images of comets (see Fig. 4d,e; new Fig,4d and Fig. 6e,f) tails, but also included a digital image from program software where the tail is seen in orange. Included in new Fig, S5) are commercial standards run side by side to validate the comet assay (new Fig. S5a,b). Gel images are shown in Fig. 4d,e, and digital images are shown in Fig. 4f. We have calculated the comets as tail length, % tail, and tail moment (Fig. S5d). All measurements agree,

(5) *P. 12/lines 280-281: It is very important to state the range of observed transfection frequencies observed in the experiments.*

A strength of the landscape analysis is that all the plasmids are transfected together in the same reaction. Primary glia was transfected with sufficient efficiency (4-8%) to afford a robust analysis of DNA repair capacity. The repair activity of the individual plasmids is normalized to signals from plasmids that do not contain a lesion. The assay is well established, and details of the analysis accuracy are laid out in previous articles.

- Nagel, Z. D. *et al.* Multiplexed DNA repair assays for multiple lesions and multiple doses via transcription inhibition and transcriptional mutagenesis. *Proc Natl Acad Sci U S A* **111**, E1823-1832 (2014).
- Piatt CG, Pecen TJ, Lavery DJ, Nagel ZD. Large-scale preparation of fluorescence multiplex host cell reactivation (FM-HCR) reporters. *Nat Protoc.* 2021 Sep;16(9):4265-4298.
- Nagel ZD, Beharry AA, Mazzucato P, Kitange GJ, Sarkaria JN, Kool ET, Samson LD. Fluorescent reporter assays provide direct, accurate, quantitative measurements of MGMT status in human cells. *PLoS One.* 2019 Feb 27;14(2):e0208341.

(6) *Fig. 5d/Line 290-291: What constitutes a “trend”? It looks to me like the relative activities of different pathways as measured by this assay are very similar in the 4 sampled populations. But that’s as likely to be a by-product of isolation and culturing, so it doesn’t merit saying much more, in my opinion.*

We did not see consistent passage-dependent effects on repair activity, but some variability was introduced in the biological replicates, all of which were isolated from a different set of animals. However, old Panel 5c showed only that any given DNA repair pathway had about the same activity in all four brain regions. This plot was not informative and distracted from major point about the relative efficiencies of the pathways in each region and it has been eliminated in the revision. We have revised the manuscript with a new Fig. 5c where we plot the same data side by side as relative repair pathway activity per region. As is obvious by the log₁₀ scale axis, the pathway activity is significantly different in each brain region: BER activity is high and DSB activity is low in all regions.

(7) *Fig. 6 & associated text: SSB are a part of normal cell function, which ought to be noted here.*

Yes, good point--Noted in the revised manuscript.

(8) *Are the SSB numbers corrected for DSB? That seems appropriate so that you are not counting things twice. If you didn’t do that, does applying such a correction, say $NetSSB = ObsSSB - 2(ObsDSB)$, change the conclusions?*

This is a great idea (and what we would have liked to do). However, given the difference in gel running conditions between neutral and alkaline comet, there are not yet robust and accurate methods to relate the two by this type of subtraction. The firm guideline, however, is that the number of SSBs is generally a couple of orders of magnitude higher than the number of DSBs. So, the contribution of DSBs to the alkaline comet signal is minimal.

Ge J, Prasongtanakij S, Wood DK, Weingeist DM, Fessler J, Navasummrit P, Ruchirawat M, Engelward BP. CometChip: a high-throughput 96-well platform for measuring DNA damage in microarrayed human cells. J Vis Exp. 2014 Oct 18;(92): e50607.

(9) *Lines 329-330: As noted, oxidative damage includes many things, and the same processes generate other DNA damage that could lead to DSB. Free radicals hit many other targets, too, so the effect need not even depend on BER. Transcription in particular is a significant source of SSB (Alt & Schwer DNAR 71:158; 2018).*

Our model does not depend on BER being the only source of SSBs, nor does our model depend on an SSB-DSB conversion being the only source of DSBs. The model depends on an equilibrium between SSBs and DSBs and multiple pathways can contribute; BER is just the most prominent. As discussed by Alt and more recently by Milano (2024), open DNA states such as during transcription are more easily oxidized and are often sources of SSBs after base excision (*Milano et al., Mol. Cell, 2024*). DSBs can also arise by direct oxidation, but less frequently than base excision generates a SSB. All of these processes contribute to the SSB-DSB equilibrium, but some are more important sources than others. In the revision, we have included a more in-depth discussion of how multiple relevant pathways form SSBs or DSBs in response to oxidation and their likely importance.

(10) *Discussion: Does the proposed BER/DSBR imbalance predict that glia and neurons are less sensitive than other cell types to the DSB-triggered checkpoints? That might have implications as much for cancer etiology as for treatment.*

Dividing cells rely on cell cycle check points to repair DSBs at the replication fork. Adult neurons are almost exclusively post-mitotic (non-dividing) and do not have cell cycle check points. We propose that a dynamic equilibrium provides an alternative mechanism to deal with damage in the absence of replication. As we discuss in revised text, in the new discussion and new Figure 8, the SSB-DSBs equilibrium can be disturbed if oxidation is too high and pushes the equilibrium to the right. In this case, persistent SSBs might lead to somatic mutation over time, which may have implications for cancer etiology,

(11) *Lines 367-368: "If and how a DSB formation-suppression mechanism might occur" is kind of central to the thrust of the paper. But it's hard to discern a regulatory mechanism in the text, and it is not known whether the differences are causes or effects.*

We have included new experiments to support the SSB to DSB conversion model (see new Figs. 6,7,8). The new experiments (in new Fig. 7) together with the XJB-5-131 in vivo reversibility (Fig. 6) provide strong evidence for a causative effect. We have significantly revised the discussion to address a likely mechanism based on our own data together with those of others.

(11) *Line 402: Among the practical considerations is that the primary cell cultures were (apparently) not synchronized. So, the transfection efficiency, the location of the probe molecule within the cell, and the activity of different DNA repair pathways are all likely to vary in an uncontrolled fashion and substantially increase the resulting noise.*

See (5) above. Primary glia are very slow growing cells. In the landscape analysis, all the plasmids are transfected together in the same reaction and the repair activity of the individual plasmids is normalized to signals from plasmids that do not contain a lesion (see Nagal, et al., *PNAS*, 2014 and Piatt et al., *Nat. Protocol*, 2021). Indeed, as indicated by Fig. 5c, the relative activities were robust. As is obvious by the log10 scale axis, the pathway activity is significantly different but consistent in each brain region: BER activity is high and DSBR activity consistently low in all regions.

- Nagel, Z. D. *et al.* Multiplexed DNA repair assays for multiple lesions and multiple doses via transcription inhibition and transcriptional mutagenesis. *Proc Natl Acad Sci U S A* **111**, E1823-1832 (2014).
- Pieltz CG, Pecun TJ, Laverty DJ, Nagel ZD. Large-scale preparation of fluorescence multiplex host cell reactivation (FM-HCR) reporters. *Nat Protoc.* 2021 Sep;16(9):4265-4298.
- Nagel ZD, Beharry AA, Mazzucato P, Kitange GJ, Sarkaria JN, Kool ET, Samson LD. Fluorescent reporter assays provide direct, accurate, quantitative measurements of MGMT status in human cells. *PLoS One.* 2019 Feb 27;14(2):e0208341.

(12) *Lines 409-411: Note that the generation of some endogenously-generated DNA lesions is likely to vary quite a bit among cell types and tissues, with further variations arising from the physiology-dependent pathways (e.g., free radicals).*

We agree that the levels and pathways of oxidation are likely to vary among tissue. Therefore, as we see among brain regions, we fully expect tissues can vary in the extent of both lesion formation and lesion repair. We would expect that variability would be an important and characteristic part of their physiology and reflected in the metabolic flux that controls ROS production.

(14) Fig. 9: I don't find this very useful; it doesn't illuminate anything not already clear in the text.

As an expert in DNA repair, new Fig. 8 is not essential. However, many non-experts expect (and perhaps need) a summary figure to consider principles raised by the data. We have modified new Fig. 8 to include transcriptional signaling as a potential regulatory point in the model and prefer to keep the Fig. 8. But if is the editor feels it does not help, we are happy to remove it.

MINOR POINTS—

(1) *General: there are quite a few small errors/typos, indicating that better proofreading was needed. The comments below include a couple of examples, but there are more.*

We have combed the manuscript and strived for more thorough editing.

(2) *Abstract, lines 32-35: This sentence is opaque, probably as a result of trying to cram too many things into it.*

We have revised and shortened the abstract to simplify and improve transparency,

(3) *Lines 65-66: This repeats a common misinterpretation. 8-oxo-dG is formed at rates similar to many other oxidative lesions, and it constitutes just a few percent of the total. It is simply better known and somewhat better understood.*

Good point. We have modified the sentence to read “a source of oxidative damage.”

(4) *Results, lines 220-222: This is not a sentence.*

Corrected

- (5) *Lines 321-322: The sentence could be misinterpreted to suggest that you to mean that certain genome locations always have an SSB. If you mean that, say so, otherwise revise.*

We have modified the sentence to avoid confusion. "The level of 8-oxo-G/ μ gDNA was robust between 7wks and 75wks indicating that each brain region had a constitutive pool of BER substrates as a source for SSBs"

- (6) Discussion, final paragraph: Many other tissues are also relatively quiescent, so would the same process as proposed for neurons also occur in those? What evidence might there be for that?

We have not measured other tissues and, therefore, did not choose to speculate on them. However, we favor the idea that, if the tissue is truly quiescent, the same principles would hold. This has been born out experimentally in our own glial cells. Another example: (Sweigert, S. E., Eguchi-Kasai, K., Warters, R. L. & Dethlefsen, L. A. *Repair of DNA Single- and Double-strand Breaks in Proliferating and Quiescent Murine Tumor Cells. International Journal of Radiation Biology* **56**, 253-264 (1989)). A generalized process, however, brings up important caveats. (1) Many human tissues in adults are quiescent but restore themselves through differentiation during life. We know nothing about the impact of differentiating states on repair under basal conditions. (2) Tissues have different physiology. They may express different levels of repair machinery, may generate radicals at different rates, or may have distinct repair efficiencies, which would require experimental testing.

REVIEWER #3

- (1) *The study's authors demonstrate that DNA repair mechanisms are region-specific in the brain by detecting the expression of specific critical proteins related to DNA damage repair pathways through immunoblotting (Fig. 1 and Fig. S1). However, the figures only provide data from two mice (n=2) and do not indicate the intrinsic reference protein used in the assay. Therefore, it is necessary to present multiple samples (n>3) to validate the results.*

There were no instances where the samples were measured less than 3 times, All the WB data was derived from samples comprising 3-6 pooled animals which reflect the average of 3-6 biological replicates in each sample. We made two such pooled samples (hence n=2), and they were analyzed in triplicate. The SDS-PAGE gel for one of the triplicates is included in Fig. 1. The methods and legends have been checked for accuracy.

- (2) *The authors also conclude that the expression of proteins related to the DNA damage repair pathway is significantly higher in neurons than in glial cells in the same mouse brain region. However, the immunofluorescence figures shown in the manuscript lack co-staining results of DNA damage repair proteins and NeuN, which is crucial in determining the cell type of the displayed cells.*

Neurons were always identified with NeuN staining, which was used throughout as our baseline. NeuN staining was not always shown because they were difficult to resolve in 3-color overlap images (DAPI, NeuN and protein). However, in the revision, we have included new Fig. S2a to explicitly highlight co-staining of neuron specific NeuN and MSH6 in all four brain regions (CTX, CBL, STR and HIP). (b) Little staining occurs in NeuN negative glia in all four regions. To make overlap images clear (see Fig. S2a, leftmost panels), we included staining images of (DAPI, Protein, NeuN) in 3 separate color channels in addition to the overlay in panel 1 (D/N/M). In Fig. S2(c, d) we also show in the CTX that the same pattern was observed for all five repair proteins

(Fig. S2c,d): they were expressed strongly in NeuN positive neurons and weakly stained NeuN negative glia.

- (3) *The authors conclude that the primary method of DNA damage repair in the cerebellum and hippocampus is non-homologous end joining (NHEJ), based on the immunofluorescence staining results of 53BP1 in neurons. To support this conclusion, the authors must provide immunofluorescence figures of other DNA damage repair pathway proteins.*

We do not understand the comment. We did not conclude that there is a primary DNA damage repair pathway in the cerebellum and hippocampus. We show that it is the interplay of and balance between BER (lesion formation) and DSBR (lesion repair) that governs the level of residual DBSs. We show that all DNA repair pathways are present (Fig, 3) and all are active (Fig, 5) in all four brain regions. In activity assays, BER is the most active and HR/NHEJ has the lowest activity. Of the DSBR pathways, BP53 staining indicates that NHEJ may be preferred, but our conclusions do not depend on it. Expression for all five pathway proteins was measured using IF in Fig. 3. Expression levels vary in the order of APE1(BER), XPA (NER), MRE11(HR), Ku80(NHEJ), and MSH6 (MMR)(see Fig 3).

- (4) *In Fig. 4a, the authors show that γ H2AX staining is region and cell-specific in the brain. However, the immunofluorescence picture only shows one brain region, and this set of figures does not provide co-staining results with markers of neurons or glia.*

See 2 above. Additionally, we have revised Fig. 4a to include co-staining images of NeuN and γ H2A-X for MSH3 in all four brain regions and staining all five pathway proteins in the CTX. Additionally, in new Fig. 4c,d, we have included neutral comet assays in all four regions of the brain to support the γ H2A-X staining results. Both the γ H2A-X staining and neutral comet validate an age dependent increase in DSBs.

- (5) *The authors address that the accumulation of reactive oxygen species is the main reason for the formation of DNA double-strand breaks (DSBs) in the brain (Fig. 8). Nevertheless, the authors believe that DSBs in normal brain tissue are transformed from single-strand breaks (SSBs). This conclusion cannot be approved based solely on the antioxidant drug XJB-5-131 injection assay.*

The reviewer makes a good point. Therefore, we have included three additional experiments and include them in the revised manuscript together with a new Fig.8 to support a SSB to DSB conversion mechanism.

Three additional experiments provide evidence for SSB conversion to DSBs.

We have substantially revised the manuscript by inclusion of three new key experiments (in new Fig. 7a-j) that support the SSB to DSB conversion mechanism as shown *in vivo* (Fig. 6g-i). We emphasize that XJB-5-131 is a powerful proof of principle for SSB to DSB conversion *in vivo* (in Fig. 6i). Oxidation damage is repaired primarily by BER through a transient SSB intermediate. XJB-5-131 suppresses ROS and base oxidation. Since XJB-5-131 suppresses DSBs, they must also depend on oxidation *in vivo*. We found it difficult to interpret the results in any other way than the DSBs arose from two closely spaced SSBs.

Three additional experiments in cells further demonstrate SSB-DSB conversion and are included in the revised manuscript together in a new Fig. 7.

- (1) In the first analysis (new Fig. 7, top panel), we prepared confluent cultures of immortalized NIH3T3 cells and treated them with peroxide to elevate ROS in the presence or absence of the ROS inhibitor XJB-5-131. If SSB to DSB conversions depended on the oxidation state, we expected that peroxide treatment would elevate DSBs, and that the rise would be suppressed in the presence of XJB-5-131. Indeed, although peroxide induces modification of single bases, the transfected cells stained robustly with γ H2A-X and 53BP1 foci evident within 30 minutes of treatment. The DSB foci disappeared with XJB-5-131 treatment. Thus, DSB formation depended on base oxidation *in vitro* and appeared to arise from conversion of two SSBs that formed during removal of the oxidized bases.
- (2) In a second experiment (new Fig. 7, middle panel), we tested whether the reduction in DSBs by XJB-5-131 could occur in the absence of base oxidation (Fig. 7d). The premise being that SSB to DSB conversions would be overestimated if non-oxidative lesions contributed to the population of DSBs. To test this idea, cell cultures were prepared as in experiment 1 but, in this case, genomic SSBs were generated enzymatically by transient transfection of recombinant CAS9D10A. This mutant converts the Cas9 nuclease into a “nickase” capable of introducing only one SSB into DNA per reaction (*Mali, P., Yang, L., Esvelt, K. M., Aach, J., Guell, M., DiCarlo, J. E, Church, G. M. (2013). RNA-guided human genome engineering via Cas9. Science, 339(6121), 823-826.*). We targeted SSB formation to the major satellite DNA using “sense” Cas9D10A guide RNA that was complimentary to the site. In the transfected cells, we tested whether (1) SSBs could be converted to DSBs in the absence of oxidation and (2) whether DSB formation under these conditions could be suppressed by XJB-5-131. Although CAS9D10A induced only SSBs, the transfected cells stained robustly with γ H2A-X and 53BP1 markers within 6 hours post transfection. The results confirmed that SSB to DSB conversion had occurred. Furthermore, the IF signals of γ H2A-X and 53BP1 co-localized with the CAS9D10A RNA guide bound at the Major satellite loci, indicating that SSBs and DSBs formed at the same Major satellite target sites. Notably, the formation of CAS9D10-directed DSBs were not suppressed by treatment of cells with XJB-5-131. Collectively, the results from experiments 1 and 2 demonstrated that DSB formation depended on base oxidation, which was suppressed by XJB-5-131; SSB to DSB conversions occurred during excision of oxidative base damage.
- (3) In a third experiment (new Fig. 7, lower panel), we tested whether XJB-5-131 had any impact on DSB formation itself. If XJB5-131 inhibited DSBs directly, for example, then their reduction might be unrelated to oxidation. We repeated experiment 2 but replaced CasD10A with active Cas9 and targeted it to the Major satellite DNA using Cas9 guide RNA. The Cas9 nuclease cuts duplex DNA without a SSB intermediate or DNA oxidation. Thus, we expected that DSBs would occur under these conditions but independently of XJB-5-131 treatment. Indeed, DSBs formed in these cells, as detected by IF intensity of γ H2A-X and 53BP1 staining. Notably, XJB-5-131 treatment did not have an impact on formation of DSBs. Collectively, the results of the three complementary experiments provided evidence that excision of oxidized bases promoted SSB to DSB conversion, and DSBs could be modulated by regulating base oxidation *in vitro* and *in vivo*.

We emphasize that XJB-5-131 is a powerful proof of principle for SSB to DSB conversion. Oxidation damage is repaired primarily by BER through a transient SSB intermediate. XJB-5-131 suppresses ROS and base oxidation and reduces SSBs. Since XJB-5-131 suppresses DSBs, they must also depend on oxidation. It is difficult to interpret the results in any other way than the they arise from two closely spaced SSBs.

- (6) *Page 10, 2nd para, “However, the opposite was true; the level of DSB lesions (measured by γ -H2AX staining) (Fig. 4b) was directly proportional to the expression level of Ku70 and Ku80 in the same cell type (Fig. S5). No data for Ku70 is shown.*

We thank the reviewer for the comment—Ku80 was included in Fig. S5. This label has been corrected in the revision.

REVIEWER COMMENTS

Reviewer #1 (Remarks to the Author):

Polyzos et al. have now revised the manuscript and performed additional experiments. However several concerns remain and are listed below. The numbers reflect the numbering used by the authors in their responses.

Response 1a: In lines 115-116: "Although normalized to total protein for loading, the cerebellar cells were small and had five times more cells per unit total protein relative to other regions (Fig. 1c)." But, Figure 1C does not show cells per unit protein. It instead shows cells per mg tissue.

Response 1c: To be considered a biological replicate the biological sample should be processed independently. So the n is still two here and not adequate for the standard of performing a western blot. Also, the authors state that "The SDS-PAGE gel for one of the triplicates is included in Fig. 1". I don't see this in Figure 1.

Response 10: The authors performed additional experiments to show the SSB to DSB conversion (Figure 6). However, these are performed in a fibroblast cell line (NIHT3) and cannot be used to make conclusions about brain cells. Especially one of the major points of this paper is that neurons have higher levels of DNA repair even compared to glia (Line 130). The expectation was these experiments would be performed in neuronal and glial cultures, to make conclusions about DNA repair in the brain.

Additionally, some controls are missing:

-In the H2O2 experiment Figure 7 b, c. It's necessary to show the control – no H2O2 treatment.

-Similarly, please show the controls with no CasD10A and Cas9. These negative controls are important to interpret the γ H2AX signal.

Response to 12: Finally, as indicated in the first peer-review comments- these experiments even when done in brain cells only show the conversion of SSB to DSB. However, there is still no evidence to show that DSBs act as sensors for SSB as opposed to cells sensing SSBs directly through well well-studied mechanisms.

Response to 13: The statement referred to here was not made by this reviewer (Reviewer 1)

Reviewer #2 (Remarks to the Author):

The authors have made an effort to address the critiques of their initial manuscript. While that effort addressed primarily technical issues, and the writing of the paper is improved, I am left wondering how much of the observed effects are the results of "regulation", versus how much is merely a consequence of the amount of repair activity a given cell type for SSB vs. DSB. As noted by the authors, there are already published results identifying regions such as enhancers as hotspots for SSB and associated DNA repair, and they touch on the

possibility that transcription might be a significant part of the SSB/DSB numbers. But this may all be a passive consequence of the transcription, DNA damage, and DNA repair processes occurring at the same time.

Reviewer #3 (Remarks to the Author):

The authors have addressed all of my concerns and I would recommend the manuscript for publication.

Reviewer 1.

Polyzos et al. have now revised the manuscript and performed additional experiments. However, several concerns remain and are listed below. The numbers reflect the numbering used by the authors in their responses.

- (1) *Response 1a: In lines 115-116: "Although normalized to total protein for loading, the cerebellar cells were small and had five times more cells per unit total protein relative to other regions (Fig. 1c)." But, Figure 1C does not show cells per unit protein. It instead shows cells per mg tissue."*

We have corrected the text to indicate cells per mg tissue.

- (2) *Response 1c: To be considered a biological replicate the biological sample should be processed independently. So the n is still two here and not adequate for the standard of performing a western blot. Also, the authors state that "The SDS-PAGE gel for one of the triplicates is included in Fig. 1". I don't see this in Figure 1.*

The legend in Fig. 1 was confusing and it has been re-written.

We evaluated 4 genetically identical animals in the clonal C57BL/6J colony, n=2 at 10 weeks and n=2 at 75 wks. The two 10-wk samples (Fig. 1a) and the two 75-wk samples (Fig. 1b) were evaluated in five gels, each probed with a single antibody. On each gel, the extracts from the two samples were run side by side, indicated by the number 1 and 2 in the gel. In all cases, the brains of all four animals expressed the machinery needed to carry out DNA repair, as follows.

(in revised legend). The C57BL/6J colony is clonal and all animals express the same protein profile. Thus, we selected four genetically identical animals (n=4) from the C57BL/6J colony, n=2 evaluated at 10 weeks and n=2 evaluated at 75 wks. Five replicate set of SDS-PAGE gels were evaluated for the two 10-wk samples (Fig. 1a) and for the two 75-wk samples (Fig. 1b), each probed with a single antibody. At both ages, the extracts from the two samples were run side by side, indicated by the number 1 and 2 in the gels. In all cases, the brains of all four animals (n=4) expressed the machinery needed to carry out DNA repair.

We wish to emphasize an important point. Although n=3 is a standard if you want to compare or quantify values, the necessary number of biological replicates is defined entirely by the experiment. (i.e., is inversely proportional to the expected differences in the outcome). In our case, for Fig. 1, we made no quantitative comparisons. As a beginning to detailed analysis, we asked only whether DNA repair proteins were or were not expressed in the brain (yes or no). The premise being that it was of value for us to know that the proteins actually existed (expressed) before launching into detailed quantitative analysis of their expression. Since the C57BL/6J animals are clonal, this type of yes-no determination in genetically identical animals formally requires no more than 2 animals; protein expression is detected in animal 1 (yes or no) and expression is reliably reproduced in genetically identical animal 2. Nonetheless, we detected protein expression in all four samples. Protein expression level was measured again in Fig. 3, where quantitative regional comparisons of the protein expression were presented in the single cell analysis.

- (3) *Response 10: The authors performed additional experiments to show the SSB to DSB conversion (Figure 6). However, these are performed in a fibroblast cell line (NIHT3) and cannot be used to make conclusions about brain cells. Especially one of the major points of this paper is that neurons have higher levels of DNA repair even compared to glia (Line 130). The expectation was these*

experiments would be performed in neuronal and glial cultures, to make conclusions about DNA repair in the brain.

We designed generic proof of principle tests, which could apply to any cell, as follows:

- We established that both brain cells *in vivo* and in NIH3T3 cells respond the same way to oxidation: peroxide treatment induced DSBs, which were reversible by XJB-5-131 indicating a SSB to DSB conversion in either cell type. Since BER induces a transient SSB during removal of oxidized bases and XJB-5-131 suppresses ROS production and single base oxidation, we found it difficult to explain the appearance of DSBs in any other way than by SSB to DSB conversion. Since neurons are difficult to culture and transfect, additional support for a SSB to DSB conversion was obtained only in NIH3T3 cells. *However, we designed generic proof of principle tests, which could apply to any cell, as follows.*
- We followed the fate of the SSB and DSBs by artificially controlling their formation. The targeted induction of SSB (Fig. 7d-f) or DSBs (Fig. 7g-i) did not arise from cell physiology, but rather were enzymatically produced by Cas9 as proof of principle. Cas10A makes SSB and Cas9 produces DSBs in any cell. Furthermore, in these tests, we followed the production of DSBs and not their repair. Therefore, the proof of principle testing did not depend on the level of expressed DNA repair proteins or regulation of DNA repair. We designed the experiment to test solely whether DSB could arise when only SSBs were generated. Thus, the proof of principle required a cell, but not a particular cell type.

In the revision, we have softened our statement. “Both brain cells and NIH3T3 cells appear capable of SSB to DSB conversions in response to oxidation (based on the reversible DSBs after XJB-5-131 treatment) and may occur by a similar mechanism.” We have eliminated any reference in the abstract, and introduction to this active process and limit reference to SSB to DSB conversion only to the last section in the results and in discussion of the model.

(4) *Additionally, some controls are missing: -In the H2O2 experiment Figure 7 b, c. It's necessary to show the control – no H2O2 treatment. -Similarly, please show the controls with no CasD10A and Cas9. These negative controls are important to interpret the γ H2AX signal.*

We have revised the paper to include no H₂O₂ control, no Cas910A and no Cas9 controls in the supplement. No DSBs are present.

(5) We have combined the response to the final comment of Rev. 1 and the only comment from Rev. 2, as they address some of the same points.

“(Rev. 1) Response to 12: Finally, as indicated in the first peer-review comments- these experiments even when done in brain cells only show the conversion of SSB to DSB. However, there is still no evidence to show that DSBs act as sensors for SSB as opposed to cells sensing SSBs directly through well well-studied mechanisms.

“(Rev 2).. The authors have made an effort to address the critiques of their initial manuscript. While that effort addressed primarily technical issues, and the writing of the paper is improved, I am left wondering how much of the observed effects are the results of “regulation”, versus how much is merely a consequence of the amount of repair activity a given cell type for SSB vs. DSB. As noted by the authors, there are already published results identifying regions such as enhancers as hotspots for SSB and associated DNA repair, and they touch on the possibility that transcription might be a

significant part of the SSB/DSB numbers. But this may all be a passive consequence of the transcription, DNA damage, and DNA repair processes occurring at the same time.

There is no disagreement. *Indeed, Rev. 1 and 2 summarize exactly features of the model we propose. We agree completely that DNA damage, DNA repair and transcription occur together in cells (Rev 2), and that SSB are constitutively sensed and repaired in brain cells by their canonical mechanisms (Rev 1). What our data show, however, is that canonical repair in brain cells is characterized by high BER and low DSBR. The interplay of these repair systems creates conditions where elevated SSBs on opposing strands can passively convert to DSBs and interconvert in response to oxidative stress. Thus, DSBs passively form in the context of a normal background of sensing, repair, and transcription, which occur together. As suggested by the reviewer, what we more accurately describe is an overarching relationship in which a cell integrates passive physiological processes to avoid runaway DSBs. It was insightful of the reviewers to raise the point of passive transitions, and we have re-written the text not only to be clearer, but also based on the reviewer's comments, we have renamed the model (in Fig.8a) as a "passible equilibrium model".*

Transcription is constitutive in any cell and in many cases is up or down regulated as needed. Thus, we have proposed a hypothetical model for a possible use of transcription in modulating SSB-DSB equilibrium. The model incorporates the known, adaptive features of DSBs that form in enhancers to transiently stimulate gene expression (see citations below). Our conclusions do not depend on the model. It is simply a possible mechanism that is consistent with the fact that (1) DSBs are elevated in animals during exercise and revert to baseline when animals are at rest, (2) DSBs induce immediate early genes, and (3) with our finding that DSBs constitute a balance, where pathways that facilitate repair are as important as those that generate the breaks.

- Pollina, E. A. *et al.* A NPAS4-NuA4 complex couples synaptic activity to DNA repair. *Nature* **614**, 732-741 (2023).
- Dileep, V, Tsai, LH. Neuronal enhancers get a break. *Neuron*. **109**, 1766-1768 (2021).
- Madabhushi, R. *et al.* Activity-Induced DNA Breaks Govern the Expression of Neuronal Early-Response Genes. *Cell* **161**, 1592-1605 (2015).
- Suberbielle, E. *et al.* Physiologic brain activity causes DNA double-strand breaks in neurons, with exacerbation by amyloid- β . *Nature neuroscience* **16**, 613-621 (2013).
- Sutormin, D. A. *et al.* Diversity and Functions of Type II Topoisomerases. *Acta Naturae* **13**, 59-75 (2021).
- Boutros, S.W, Unni, V. K. & Raber, J. An Adaptive Role for DNA Double-Strand Breaks in Hippocampus-Dependent Learning and Memory. *Int J Mol Sci* **23**, 8352 (2022).
- Konopka, A., Atkin, J.D. The Role of DNA Damage in Neural Plasticity in Physiology and

Rev 3. "The authors have addressed all of my concerns and I would recommend the manuscript for publication."

REVIEWERS' COMMENTS

Reviewer #1 (Remarks to the Author):

The authors have addressed all my comments and have generalized the language to account for non-brain cell types used in the study